# The Interfaces Twitter Elections Dataset: Construction process and characteristics of big social data during the 2022 presidential elections in Brazil

**Sylvia Iasulaitis**[1,2,3☯]*, **Alan Demétrius Baria Valejo**[4,5☯], **Bruno Cardoso Greco**[6☯], **Vinicius Gonçalves Perillo**[4☯], **Guilherme Henrique Messias**[5☯], **Isabella Vicari**[2☯], **with the Interfaces—Center for Sociopolitical Studies of Algorithms and Artificial Intelligence**

1 Department of Social Sciences, Federal University of São Carlos, São Paulo, Brazil, 2 Graduate Program in Science, Technology, and Society, Federal University of São Carlos, São Paulo, Brazil, 3 Graduate Program in Information Science, Federal University of São Carlos, São Paulo, Brazil, 4 Department of Computing, Federal University of São Carlos, São Paulo, Brazil, 5 Graduate Program in Computing Science, Federal University of São Carlos, São Paulo, Brazil, 6 Department of Information Science, Federal University of São Carlos, São Paulo, Brazil

☯ These authors contributed equally to this work.
* si@ufscar.br

**Data Availability Statement:** https://github.com/Interfaces-UFSCAR/Codigo-Coleta-PLOS-ONE and https://github.com/Interfaces-UFSCAR/ITED-Br.

## Abstract

The main objective of this study is to describe the process of collecting data extracted from Twitter (X) during the Brazilian presidential elections in 2022, encompassing the post-election period and the event of the attack on the buildings of the executive, legislative, and judiciary branches in January 2023. The work of collecting data took one year. Additionally, the study provides an overview of the general characteristics of the dataset created from 282 million tweets, named "The Interfaces Twitter Elections Dataset" (ITED-Br), the third most extensive dataset of tweets with political purposes. The process of collecting and creating the database for this study went through three major stages, subdivided into several processes: (1) A preliminary analysis of the platform and its operation; (2) Contextual analysis, creation of the conceptual model, and definition of Keywords and (3) Implementation of the Data Collection Strategy. Python algorithms were developed to model each primary collection type. The "token farm" algorithm, was employed to iterate over available API keys. While Twitter is generally a "public" access platform and fits into big data standards, extracting valuable information is not trivial due to the volume, speed, and heterogeneity of data. This study concludes that acquiring informational value requires expertise not only in sociopolitical areas but also in computational and informational studies, highlighting the interdisciplinary nature of such research.

## Introduction

Profound changes in the information society have led to a sharp increase in the volume of data available on the Internet. Since 2010, digital content on the Web has grown exponentially,

**Funding:** This work was funded by FAPESP under grant number 2022/03090-0 under the coordination of Prof. Dr. Sylvia Iasulaitis and BCO number 2023/03704-0 and number 2023/17214-5. This research was supported by the Coordination for the Improvement of Higher Education Personnel - Brazil (CAPES) - Finance Code 001, through the granting of a master's scholarship. The funders had no involvement in the study design, data collection and analysis, decision to publish, or preparation of the manuscript.

**Competing interests:** The authors declared that there are no competing interests.

rising from 2 Zetabytes in 2010 to approximately 97 Zetabytes in 2022 [1], with a forecast of exceeding 180 Zetabytes in 2025. This indicates a trend of doubling the amount of online data over the next two years.

The widespread use of computers—including desktops, notebooks, and, especially, smartphones—has not only facilitated the consumption but also the massive production of digital content by internet users, particularly through interactions on social media platforms.

The term "big data" emerges in this context to define the massive amounts of digital data produced. Tsai et al. [2] identify speed, volume, and variety as the main components of big data. In this study, big data alludes to various facets of the process of extracting informational value and sheds light on the challenges of identification, collection, storage, analysis, and visualization of large volumes of data.

Among big data sources, social media platforms such as Facebook, Instagram, and Twitter (renamed as "X" in July 2023) stand out. Users create accounts or profiles on these platforms, share information, and connect with others by following their respective accounts.

The fundamental conception underlying social network theory is that apparently autonomous individuals and organizations are, in reality, embedded in social relationships and interactions [3].

Pew Research reports have extensively documented how the emergence of social media platforms has influenced communication patterns worldwide, work and consumption behaviors, communities, and how individuals obtain and share information about health, politics, civic life, relationships, and even people's stress levels [4].

In politics, social media has proven to be an essential tools in electoral campaigns, aiding in building public image and support networks for politicians. Simultaneously, it is utilized to deconstruct opponents' images through negative campaigns, dissemination of fake news, and hate messages, including the use of bots.

Social media platforms played a prominent role in significant recent events, including the 2016 US presidential campaign of Republican candidate Donald Trump, the Brexit referendum in the same year leading the UK to initiate the process of leaving the EU, and Jair Bolsonaro's 2018 presidential campaign in Brazil.

Therefore, data from these platforms have garnered the attention of researchers from various fields of knowledge. Social media data analysis can offer insights into human behavior and interaction, contributing to a deeper understanding of public opinion on specific topics. It aids in identifying population niches, studying group changes over time, pinpointing influential social agents, and even developing strategies for product or service recommendations [5].

The process of gathering this data is known as data scraping, which involves searching for hidden information within a large dataset using algorithms [6]. In addition to data scraping, data mining and information retrieval are employed to manage, process, analyze, and visualize the extensive amount of structured or unstructured big data. This field has experienced exponential growth and has become increasingly institutionalized in the 21st century [7].

In this context, the application programming interface (API) technique has been widely employed to extract large amounts of data from social media platforms (referred to as big social data) [8–10], as was the case in this research.

However, it is worth noting that the structure of APIs used by major social media platforms like Twitter (X) and Facebook imposes significant restrictions on research possibilities, tool usage, and the types of data that can be collected [11].

The main objective of this study is to describe the process of collecting data extracted from Twitter (X) during the Brazilian presidential elections in 2022, encompassing the post-election period and the event of the attack on the buildings of the executive, legislative, and judiciary branches in January 2023. Additionally, the study provides an overview of the general

characteristics of the dataset created from 282 million tweets, named "The Interfaces Twitter Elections Dataset" (ITED-Br). The acronym ITED-Br alludes to the research group responsible for its creation, namely, "Interfaces—Center for Sociopolitical Studies of Algorithms and Artificial Intelligence.

This article is structured as follows:

1. The literature review section identifies available datasets and corpora containing data from discussions regarding political events on Twitter (X). It reviews articles that work with datasets featuring the most extensive corpus of political data, positioning our dataset within this literature;

2. In the materials and methods section, the challenges encountered in obtaining data based on the use of the Twitter (X) platform API are detailed, along with the strategies adopted in the face of the imposed restrictions. Additionally, this section presents the study flowchart;

3. Subsequently, the results of the study are presented, featuring an exploratory analysis of the corpus, descriptive measures of the dataset, and sample analyses;

4. The next section presents the discussion of results;

5. The Conclusion addresses the challenges of creating a database of this magnitude, especially in the context of collecting, storing, and using big social data. It also offers suggestions for future research and potential uses of the database.

The main contributions of this article consist of:

- Describing the challenge of collecting, storing, and utilizing big social data, particularly in the context of scientific research.;

- Presenting and analyzing data on a relevant topic, specifically the use of social media platforms in one of Brazil's most polarized electoral campaigns and one of the most polarized globally. The second round of this election featured Luiz Inácio Lula da Silva (Workers Party —PT, left-wing) and Jair Messias Bolsonaro (Liberal Party—PL, far-right wing);

- Providing relevant data for studies on sociopolitical phenomena and dynamics within social media platforms;

- Making the third-largest corpus of political data from social media available for areas that require large volumes of real or informationally-rich data, which does not always apply to synthetic datasets. The database will be made available on GitHub.

Making it available means offering access as conveniently as possible without losing data or important data characteristics.

This work was motivated by the substantial amount of data often existing in a 'public' access environment regarding social processes or dynamics. However, these data are not necessarily available in a form that can be easily understood or from which value can be extracted by those who could, and often should, benefit from such information.

## Literature review

The literature review aimed to identify previously published datasets and corpora containing data from discussions about political events on Twitter, along with articles featuring extensive big social data corpora found in the literature.

The search was conducted in the main collection of the Web of Science database, considering all publications—scientific articles, book chapters, and works presented at events—

published between January 1, 2010, and June 1, 2023. The search focused on studies about the use of Twitter (X) for political purposes, analyzing messages extracted from the platform, commonly referred to as "tweets." The keywords "Twitter" must appear in the text, along with the terms "politics" or "election" and "corpus" or "dataset."

The identified studies had diverse objectives, including analyzing the population's perception of political events, examining the use of Twitter by political figures and/or parties, detecting bots, fake profiles, or spam on Twitter, identifying misinformation or hate speech in tweets, predicting the results of an election, analyzing relationships between traditional media and Twitter, proposing a method or model for analyzing or collecting tweets and presenting and/or discussing a dataset/corpus. However, this review will focus specifically on works featuring an extensive corpus and articles to offer a detailed dataset, making it publicly available for future research.

Thus, only 13 articles out of the 153 found in the search met the criteria of having an extensive corpus and a detailed dataset and were consequently selected.

Regarding the **language** of the available datasets, 6 have tweets in English; 1 in Spanish; 1 in Italian; 1 in German; 1 in Arabic; 1 in English and Hindi; 1 in English and Pidgin, and 1 dataset was multilingual.

Table 1 displays the 13 corpora from the selected articles in this literature review, with the dataset built in this study positioned among them in **order of size**:

The largest corpus was published by Chen, Deb, and Ferrara [12], and it is the first public dataset with tweets from the 2020 United States presidential elections, available on GitHub. The collection of tweets began in May 2019 and covers the Republican and Democratic primaries; the confirmation of Donald Trump and Joe Biden as candidates for their respective parties; the controversy regarding the postal vote system introduced as a result of the COVID-19 pandemic; confirmation of Joe Biden's victory; the recurring allegations made by Donald Trump that the election had been rigged and the other events that culminated in the invasion of the Capitol on January 6, 2021. As of January 22, 2021, the dataset had 1,258,209,617 tweets in several languages, and the authors intended for the collection to extend to the first six months of Joe Biden's administration.

As this dataset focuses on the US elections, the predominant language of the tweets is English. The dataset also includes data such as the tweet's publication date, information about the author, and whether the tweet was original or a reply, retweet, or quote to another tweet. In addition to the data on tweets and users, the authors observed that less than 1% of tweets have information about the author's location and developed their own technique for identifying location, described in the article. Finally, one of the limitations of the dataset is the collection format allowed when the work began. At that time, the Twitter API permitted access to approximately 1% of the flow of all tweets in real-time and returned those that had any of the keywords pre-established. However, in 2021, Twitter launched a license for academic research, which allows researchers full access to the files found in the search.

The second-largest corpus was presented by the research of Kandasamy et al. [13]. Faced with one of the challenges posed by the COVID-19 pandemic, they collected 312 million tweets in English about the pandemic and proposed a method for analyzing sentiments. The authors were based on the enormous flow of information on social media, accompanied by an erroneous orientation that resulted in complications for the health sectors of governments in several countries. The research method employs the N-gram autoencoder integrated into a machine learning architecture. Using four classification algorithms—Decision Tree, Support Vector Machine (SVM), Random Forest, and K-Nearest Neighbors (KNN)—the accuracy of the sentiment analysis approach was 87.75%.

**Table 1. Corpora from the articles in order of size.**

| No | Title | Dataset Name | Authors | Corpus | Year | Language |
|---|---|---|---|---|---|---|
| 1 | #Election2020: the first public Twitter dataset on the 2020 US Presidential election | #Election2020 | Emily Chen et al. | 1.2 billion | 2022 | Multilingual (English) |
| 2 | Sentimental Analysis of COVID-19 Related Messages in Social Networks by Involving an N-Gram Stacked Autoencoder Integrated in an Ensemble Learning Scheme | - | Venkatachalam Kandasamy et al. | 312 million | 2021 | English |
| 3 | The Interfaces Twitter Elections Dataset 2022: construction process and characteristics of big social data and the political twittersphere in Brazil | ITED-Br—Interfaces Twitter Elections Dataset | Sylvia Iasulaitis et al. | 282 million | 2022 | Portuguese |
| 4 | Influence of fake news in Twitter during the 2016 US presidential election | - | Alexandre Bovet et al. | 171 million | 2019 | English |
| 5 | NivaDuck—A Scalable Pipeline to Build a Database of Political Twitter Handles for India and the United States | PoliTwictionary | Anmol Panda et al. | 120 million | 2020 | English and Hindi |
| 6 | Twitter, Public Opinion, and the 2011 Nigerian Presidential Election | - | Clay Fink et al. | 107 million | 2013 | English and Pidgin (Nigerian English) |
| 7 | Predicting Twitter Users&x2019; Political Orientation: An Application to the Italian Political Scenario | - | Matteo Cardaioli et al. | 9.5 million | 2020 | Italian |
| 8 | Twitter social bots: The 2019 Spanish general election data | - | Javier Pastor-Galindo et al. | 5.8 million | 2020 | Spanish |
| 9 | The EPINetz Twitter Politicians Dataset 2021 | EPINetz Twitter Politicians Dataset | Tim König et al. | 426.614 | 2022 | German |
| 10 | Measuring Extremism: Validating an Alt-Right Twitter Accounts Dataset | Alt-Right Twitter Accounts Dataset | Joshua Thorburn et al. | 123.295 | 2018 | English |
| 11 | The First 100 Days: A Corpus Of Political Agendas on Twitter | The first 100 days | Nathan Green et al. | 59.789 | 2018 | English |
| 12 | SEHC: A Benchmark Setup to Identify Online Hate Speech in English | Multi-domain hate speech corpus (MHC) | Soumitra Ghosh et al. | 10.242 | 2023 | English |
| 13 | Developing A Multilabel Corpus for the Quality Assessment of Online Political Talk | Twitter Deliberative Politics dataset | Kokil Jaidka | 6.000 | 2022 | English |
| 14 | Tb-SAC: Topic-based and Sentiment Classification for Saudi Dialects Tweets | Tb-SAC | Sara Alzahrani et al. | 4.301 | 2020 | Arabic |

Note: The hyperlink at the title of each article leads to its dataset when available.

The literature review provides a framework to position the dataset presented in this study among similar ones. This dataset, named ITED-Br—Interfaces Twitter Elections Dataset 2022, gathered tweets posted during the 2022 presidential elections in Brazil. When considered alongside the 13 datasets identified in the literature review, ITED-BR ranked as the third most extensive dataset of tweets with political purposes. It comprises more than 280 million tweets (precisely 282,135,572) and can be regarded as the most comprehensive when evaluating datasets in languages other than English.

With the fourth largest corpus, Bovet and Makse [14] analyzed the dynamics and influence of fake news during the 2016 United States elections. In their research, the authors used a dataset containing 171 million tweets [15] collected in the five months prior to election day and identified 30 million tweets made by 2.2 million users that contained links to news outlets. As a result, 25% of these tweets spread false or biased news, and Donald Trump's supporters were responsible for influencing the dynamics of fake news spread, even though the center and left-wing news disseminators were the most influential on the platform.

Panda et al. [16] had the fifth largest corpus of the review, with around 120 million tweets. The authors' dataset, PoliTwictionary, included around 80 million tweets from Indian politicians and 40 million tweets from American politicians. The primary objective of the study was to introduce the dataset formed of tweets from political actors from India and the United

States built by NivaDuck, a two-step classification pipeline designed by the authors. NivaDuck was used to identify politicians on Twitter through the description section in their accounts and the content of their tweets. According to the researchers, it identified more than 18,500 Indian politicians and more than 8,000 American politicians, and the dataset was complemented by human verification.

Sixth, with 107 million tweets from 246,000 users, Fink et al. [17] constructed a dataset to analyze the sentiments of Nigerian Twitter users during the 2011 elections and compare the results obtained on Twitter with the results presented by electoral polls. The authors' collection spanned between April 2010 and April 2011, using Twitter's API 1.0 geographic query feature. When comparing, the authors concluded that the correlation between sentiment analysis and electoral polls was not significant and that Twitter, due to the freely discussed topics, should be used as a complement to opinion polls and not as an instrument for assessing public opinion.

The seventh largest corpus was introduced by Cardaioli et al. [18], offering a dataset consisting of 6,685 users and 9,593,055 Italian tweets that were labeled according to political orientation. The collection was carried out in two stages. The first started by identifying a thousand users who expressed their political orientations through retweets or comments on party publications, and the second was the random selection of Italian Twitter users. In addition, the authors discuss the feasibility of automatically classifying political orientation using machine learning techniques.

In the article "Twitter social bots: The 2019 Spanish general election," Pastor-Galindo et al. [19] present the eighth-largest dataset comprising 5.8 million tweets related to the 2019 elections in Spain. The authors detail the collection carried out between October 4 and November 11 using 46 hashtags and argue that one of the objectives of the work is to enable researchers from different countries to improve the stages of data collection, organization, and pre-processing, as well as processes presented in detail in the article. The work also discusses the detection, analysis, and classification of social bots on Twitter.

The ninth largest dataset was called "EPINetz Twitter Politicians," published by König et al. [20]. It gathers 426,614 tweets in German written by 2,449 German parliamentarians, ministers, state secretaries, parties, and ministries throughout 2021. The authors carried out an exploratory analysis to demonstrate the applicability of the dataset during the 2021 German federal elections, in addition to expressing the intention to update the corpus annually for possible longitudinal analyses.

The work of Thorburn, Torregrosa, and Panizo [21] introduces the Alt-Right Twitter Accounts dataset, ranking as the tenth largest. It comprises 422 Twitter users associated with the "Unite the Right" rally, an extremist movement originating in Charlottesville, United States, in 2017. The dataset contains 123,295 tweets, and the authors believe that these messages will contribute to studies on radicalization in online environments. Additionally, they aim to enhance understanding of the language used by the movement members.

Green and Larasati [22] published the eleventh dataset analyzed in this study, called "The First 100 Days," containing tweets in English. The collection is based on tweets made by the President of the United States and Senators in the first 100 days of his term, resulting in a corpus of 59,789 tweets. The dataset aims to identify the process of imposing agendas by parties on the new president and assist future work with a focus on linguistics and Natural Language Processing (NLP), emphasizing the relationship between language and politics.

Ghosh et al. [23] presented the twelfth dataset, composed of more than 10,000 tweets in English, called "Multi-domain hate speech corpus" (MHC). It contains hate speech messages against religion, nationality, ethnicity, and gender. The corpus, collected between January 2018 and May 2020, was manually annotated to distinguish between tweets containing or not containing hate.

Jaidka [24] published the thirteenth largest dataset, the "Twitter Deliberative Politics dataset," featuring 6,000 political tweets labeled according to their deliberative characteristics. The collection was carried out between January 2017 and March 2018, using a sample of 1% of the responses made by Twitter users to the 536 congressmen in office in the United States.

The last dataset on the list of the largest found in the literature (14th presented in Table 1) was published by Alzahrani et al. [25]. The authors consolidated efforts to create a corpus of 4,301 tweets in Saudi dialects called "Tb-SAC." The corpus was labeled by the authors based on the scale between positive, negative, or neutral tweets, and the article details the data cleaning and pre-processing steps.

## Materials and methods

The process of collecting and creating the database for this study went through three major stages, subdivided into several processes, as demonstrated in the flowchart presented in Fig 1. A preliminary analysis of the platform and its operation was carried out, as described below:

A preliminary analysis of the platform and its operation was conducted as described below:

### Twitter (X)

Twitter is a microblogging social media platform that, until February 8, 2023, allowed users to write posts of up to 280 characters. From this date onwards, the system began allowing selected users to write posts of up to 4000 characters. In any case, this study does not include tweets published after February 8, 2023.

Users interact with posts in various ways. They may express their appreciation for posts created by other users by clicking the "like" icon; they can repost (or retweet) other users' posts, write a reply to a tweet written by others, or quote others' tweets, giving relevance to that post on the platform. Furthermore, users can use hashtags to index their posts, which are included in the body of the text.

**Likes.** A "like" is a simple interaction that helps boost a post or tweet across the platform, increasing its relevance. The platform's algorithm tends to present content similar to those the user liked in their timeline (chronological feed of tweets the user sees when they log in). However, Elon Musk's acquisition of the platform was marked by several changes, such as the creation of two feeds with different logic regarding the filtering of content and the implementation of a subscription program, where paying accounts are distributed more prominently throughout the platform.

**Repost or retweet.** One of the most common ways of spreading a post on Twitter is through reposting or retweeting. The user clicks the "repost" icon to share other users' posts with their followers. Although it appears in the reposting user's timeline, the platform attributes all interactions with the reposted tweet to its source. Therefore, the reposting user loses the information chain regarding their followers' interactions with the reposted content.

**Quote.** Quoting is very similar to reposting. The difference is that now the user produces content while referencing the original tweet. Therefore, the platform considers this action a new tweet, and the interactions are attributed to the new content. This means that the user does not lose the information chain.

**Reply.** A reply consists of a tweet in response to another. This interaction involves the production of new content. Thus, it is considered a new post, subject to its own interactions. It is possible to create a chain of replies that are stored by the platform, with interactions that can be retrieved.

**Hashtags.** Twitter uses hashtags as a unique structure to group content. Users mark tweets with keywords preceded by a hashtag so the system recognizes tweets addressing a

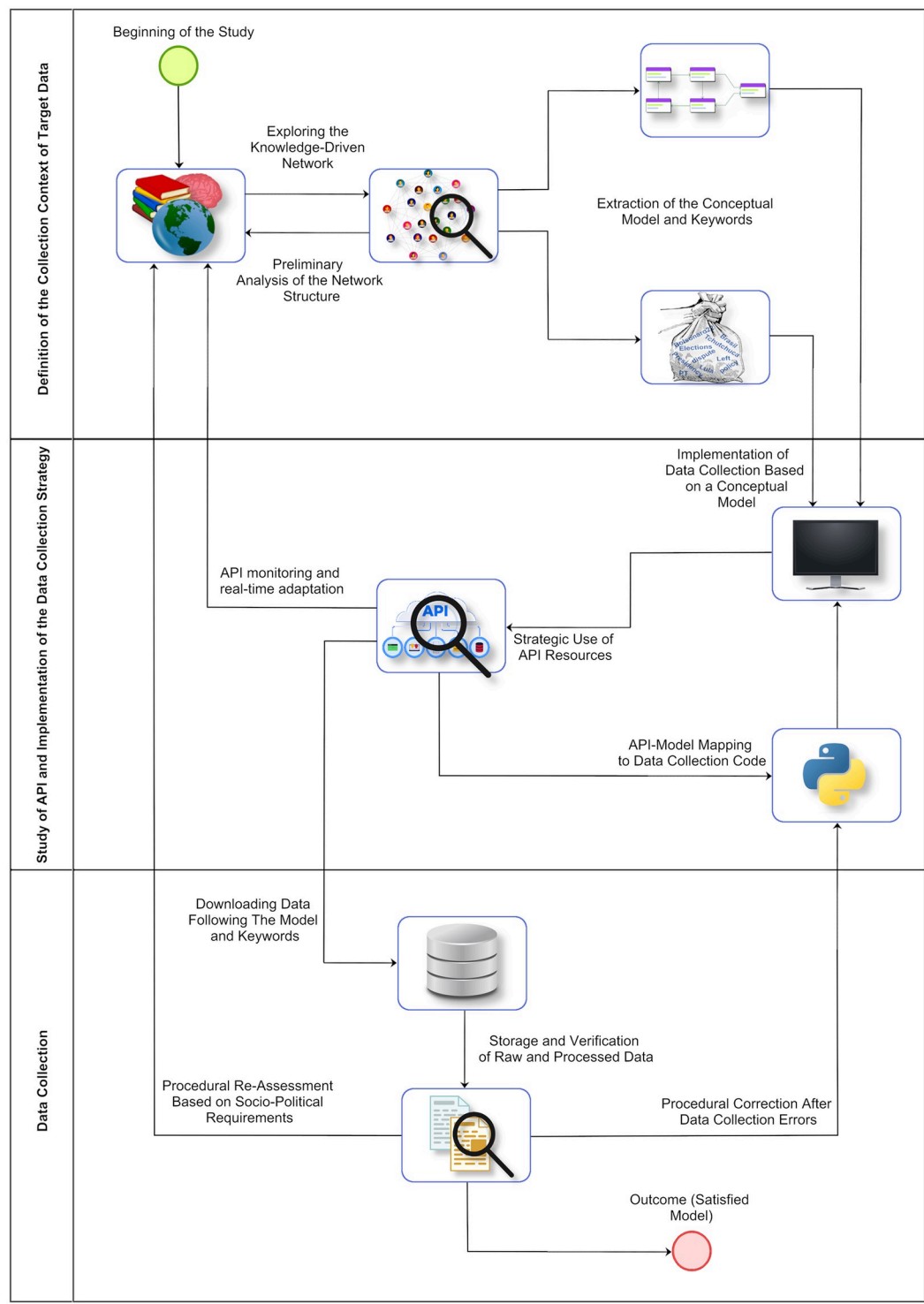

**Fig 1. Methodological flowchart.** The study's flowchart.

specific topic. The platform compiles tweets using the same marked keywords, indicating to the Twitter community the trending topics discussed on the platform.

This study used the Twitter API to collect data, which is available to the public but with some restrictions that will be explained later.

**Twitter API.** Twitter's application programming interface (API) was considered sufficient and appropriate for data collection. The Twitter API was chosen over scraping data from the website's HTML pages due to the ease and structure it provides. In theory, data scraping offers more substantial control over what (and how much) data would be obtained during collection. However, implementing several mining strategies and automatically identifying the data fields could add significant complexity to the study. Furthermore, the Twitter API offered numerous data fields and collection configurations, so it had already modeled almost all attributes considered at the beginning of the research—or the granularity offered made it simple to construct these attributes from the collectible fields.

## Contextual analysis, creation of the conceptual model, and definition of keywords

Data of interest for the collection were identified through meta-analysis and conceptual exploration using social network analysis models. Additionally, a literature review was conducted to identify studies that could contribute to meeting informational requirements. In this manner, actors, attributes, and relationships with different priority levels for the collection were identified.

Based on the conceptual model created, the simultaneous implementation of three types of collection would satisfy informational requirements: (1) by Query (term, search phrase, or hashtag), (2) by user, and (3) by tweet (tweet, quote, or repost/retweet)..

During the collection period, certain terms, hashtags, or actors (egos) were chosen as the basis for the process. Initially, some data collection procedures were modeled for exploratory purposes. From the beginning, it became clear that the volume of data would be massive, and some collections would reach the limits of the API keys individually. Therefore, it was necessary to establish priorities, as the full potential scope of collection would easily exceed the capacity of the resources available for research. In this way, the scopes determined in the analysis phase were constantly reevaluated, taking into account the daily context of Brazil's political and electoral situation during the data collection period.

**Search contexts.** The search contexts were developed by selecting prominent potential candidates during the pre-election period. We observed polls published by the companies Genial/Quaest, which interviewed 2000 voters between June 29 and July 02, 2022. The six potential candidates that stood out in terms of vote intention in the first round were Luiz Inácio Lula da Silva (Workers Party—PT) with 45% of the vote intentions, followed by the incumbent President Jair Messias Bolsonaro (Liberal Party—PL) with 31%, Ciro Ferreira Gomes (Democratic Labor Party—PDT) with 6%, André Janones (Avante) and Simone Tebet (Brazilian Democratic Movement—MDB) with 2%, and Pablo Marçal (Republican Party of the Social Order—Pros) with 1%. The error margin was 2 percentage points [26].

Based on voting intention surveys, candidates Bolsonaro, Lula, and Ciro were initially chosen as research objects. However, candidate Simone Tebet had a notable participation in the first televised debate, obtaining a prominent position on social media from then on and gaining our attention as an object of research. Thus, posts about her were retroactively collected, gathering data produced in the same time frame as the other candidates. The decision to include her among the studied candidates proved to be the right one, considering her electoral

performance, which was better than Ciro's (she got third place in the number of votes in the first round).

After choosing the candidates, we collected both posts users wrote about the candidates and posts published in the candidates' accounts. In the first case, we retrieved all tweets written about the four candidates during the time frame stipulated for the research. The search queries included the candidates' names and most used nicknames, both pejorative and appreciative. The search queries for each candidate were:

- **Bolsonaro**: (#bolsonaro OR jair OR bolsonaro OR bozo OR biroliro OR "tchutchuca do central" OR bonoro OR capitao OR genocida OR mito OR bolsomito OR bolsolixo OR bolsotrump OR messias OR patriota OR b22 OR b17 OR brocha OR imbrochavel OR maçonaro) lang:pt -is:retweet

- **Lula**: (#lula OR lula OR "ex presidiario" OR lulalivre OR "9 dedos" OR luladrao OR lulaladrao OR lulinha OR nine OR luis inacio OR cachaceiro OR "sapo barbudo" OR lulao OR l13 OR "faz o L" ORlulindo OR metalurgico OR lulalkimin) -loud lang:pt -is:retweet

- **Ciro Gomes**: (#ciro OR ciro OR c12 OR cirogomes OR "ciro gomes" OR "correu pra Paris" OR bolsolula OR ciranha) lang:pt -is:retweet

- **Simone**: (#simone OR "simone tebet" OR simonetebet OR tebet OR "simone tablet" OR estepe OR s15) lang:pt -is:retweet

The query for the incumbent president and candidate for re-election Bolsonaro was changed throughout the period, including the words "brocha" and "imbrochável" (slang words related to being sexually impotent or the opposite of that) on September 7, 2022, and "maçonaro"—a combination of the words "maçom" (Freemason) and Bolsonaro—on October 4, 2022.

The collection of tweets published by the candidates' accounts consisted of contexts that allowed retrieving all tweets from each account.

- **Bolsonaro**: from:jairbolsonaro lang:pt

- **Lula**: from:LulaOficial lang:pt

- **Ciro Gomes**: from:cirogomes lang:pt

- **Simone Tebet**: from:simonetebetbr lang:pt

The strategy of searching by nicknames and pejorative terms proved extremely effective, as it allowed us to retrieve a significant number of tweets. Some tweets that exemplify the use of these terms include:

- @BrazilFight "Look. . . thousands of people I spoke to this past week are going to vote for Bolsonaro. . . the biggest Brazilian corrupt man with 9 fingers has his days numbered.". *Explanation:* The then-candidate Lula was a lathe operator who lost the little finger on his left hand in 1964.

- @BlogdoNoblat "Your question is wrong. Should the Electoral Court grant registration to an ex-convict, *cachaceiro*, gang member, mafia member, corrupt, and partner of the Primeiro Comando da Capital like Lula?" *Explanation* "*Cachaceiro*" refers to "drunkard" describing someone who habitually drinks *cachaça*, a sugarcane spirit produced in Brazil.

- @FriedHardt "The *beiçola* is so narrow-minded that he thinks there is a grudge against Bolsonaro. We have a grudge against fennel candy, and liver steak. . . As for Bolsolixo, it's something else. . ." *Explanation:* "Beiçola" or "big-lips" refers to someone not identified, probably

reputed by the tweet's author as a person who speaks too much. "Bolsolixo" is a combination of the name "Bolsonaro" and the word "trash."

- Guys, don't call Bolsonaro **tchutchucaDoCentrao**, he doesn't like it. I repeat: don't share-**tchutchuca do centrao** or tag @jairbolsonaro ok? *Explanation:* The term "tchutchuca do centrao" suggests an insult with sexual connotations, implying that Bolsonaro was submissive to centrist parties known for making behind-the-scenes deals to benefit their members and lobbyists. This term caught the attention of the international press and was translated into several languages: in French and Spanish, it was translated closer to "prostitute of the center" ("putain du centrao" and "perrita del centrao"), while in English, the press used terms like "punk," "lap dog," "darling," "bum," among others.

**Period of analysis.**   The work of collecting data took one year. The tweets collected were published from July 20, 2022, to February 23, 2023, totaling 217 days. This period was chosen due to its importance in the context of the presidential election calendar in Brazil.

The first idea was the application of queries to collect tweets created from July 20, 2022, to November 7, 2022. However, with the repercussions of the election results, we decided to extend the collection using the same type of queries to encompass tweets created until January 31, 2023. Also, new search contexts were added to allow us to capture certain events. The candidates' numbers were included since we realized we were losing some tweets referring to the candidates using this information instead of their names or nicknames. These new search contexts were applied for the presidential election's second round, from October 1 to October 30, 2022:

- **Bolsonaro**: 22 lang:pt -is:retweet

- **Lula**: 13 lang:pt -is:retweet

Furthermore, the reaction of Bolsonaro's voters after his defeat caused a huge repercussion on Twitter. Therefore, a specific query was made to grasp these manifestations. The query was applied to collecting tweets published from October 31, 2022, to February 23, 2022:

- **Post-election**: ("intervenção militar" OR "intervenção federal" OR "alexandre de morais"' OR xandão OR fraude OR Venezuela OR Cuba OR urna OR urnas OR comunismo) lang:pt -is:retweet

Finally, part of Bolsonaro's voters, upset with the election results, invaded the buildings of the Executive (*Palácio do Planalto*), the Congress, and the Supreme Court, calling for a Coup d'Etat. This event had an enormous repercussion on Twitter, and a specific search query designed to capture the event collected tweets posted from December 31, 2022, to February 12, 2023:

- **Atos golpistas**: ("Festa da Selma" OR ato OR bolsonarista OR golpe OR golpista OR baderna OR extremista OR Brasília OR "três poderes" OR invasão OR "ocupar congress" OR "atos terroristas" OR manifestação OR atentado OR patriotas OR "tomada do poder" OR guerra OR "esplanada dos ministérios" OR "congresso nacional" OR "manifestantes" OR "retomada do poder") -ucrania lang:pt -is:retweet

All queries only collected tweets in Portuguese. Because of a limitation of the keys, retweets were not collected, except retweets posted by the candidates' accounts. The collection of retweets was interrupted by the closure of the Academic API, which means that the candidates' retweets about the invasion of the public buildings in January were not collected. In addition,

some contexts were used to collect tweets up to a certain period. In the case of candidates Ciro and Simone, their reposts more recent than October 27, 2022, were not collected; for candidates Lula and Bolsonaro and for the context post-election, retweets more recent than January 13, 2023, were not collected.

## Implementation of the data collection strategy

**Strategic use of API resoures.**  The data collection process was semi-automated, necessitating the incorporation of new search contexts. Python algorithms were developed to model each primary collection type. During the API assessment and tool implementation phase, it was identified that the "Tweepy" library for Python would streamline the utilization of the Twitter API while maintaining the granularity of desired data fields. Consequently, the data collection system was implemented using these technologies and could be accessed through a terminal.

The system automatically gathered tweets based on specified search contexts (e.g., hashtags from a list, queries from a list, ego networks from a user—tweets made by this user). It was designed to pause when reaching limits and resume automatically upon release. The "token farm" algorithm, elaborated in the following section, was employed to iterate over available API keys, using those not currently at their limits.

Despite being public, the API imposed stringent temporal restrictions on the volume of data collected by a user (10 million tweets per month per token). The study's intended data collection could not be accommodated under these limitations. Thus, researcher accounts with fewer restrictions were utilized, obtained through a form provided by Twitter. The study was crafted considering the API's terms of use and limitations applicable to these researcher accounts.

The collection took place through the Twitter API, accessible upon requesting keys (or tokens) with varying access levels. The academic key used in this study is freely available for researchers globally, provided they demonstrate the scientific purpose of using APIs. This key permitted the collection of up to 10 million tweets monthly, with suspension upon reaching this limit until the next month. Additionally, each API request had a maximum limit every 15 minutes, with the request halted until the end of those 15 minutes.

To maximize the database's potential, the collection involved multiple researcher accounts, totaling six users, each with individual limits. Additionally, accounts with fewer privileges were used for testing purposes and were not involved in the actual data collection.

Another challenge was the contemporaneity of the data and the API's volume limits being reset multiple times a day. As a result, the data collection was conducted in real-time throughout the period, with the option for retroactive collection if needed.

**Token farm.**  These limitations were circumvented by utilizing six academic access keys. When one key was suspended, another was activated, optimizing data collection and allowing for the gathering of more tweets per month. This strategy involved rotating keys, ensuring available keys were used while others were in a "rest" period, waiting for a new limit to be released.

**Phases of data collection.**  With the search contexts defined, the collection was automated. The algorithm ran continuously and required a restart only in the event of external issues.

The algorithm conducted daily collections, fetching content from the day before its execution. Once all tweets for that day were gathered, a JSON stored information indicating nothing else to collect. Subsequently, the algorithm began collecting tweets from previous days that had not yet been included in the collection process. This measure ensured data contemporaneity,

allowing the algorithm to seamlessly continue from where it left off. At the end of each day, the collection status, including the information of that day, the applied search context, and the phase of the collection, was stored in JSON. This ensured that the collection would restart from the point it had concluded. Additionally, the context was periodically saved in JSON if the algorithm encountered any complications and needed restarting.

The daily collection consisted of three phases. The first involved a general collection, where all tweets from a specific day were gathered within several requests, each containing a maximum of 500 posts (the API limit). Each request had a specific ID provided by the API, indicating its position in the total collection. The information from these posts, as shown in Table 1, was collected and structured in a DataFrame using the Python library pandas. Information about the authors (Table 2) and media in each post (Table 3) were also collected and stored in separate pandas DataFrames. Consequently, three data files, named "tweets," "users," and "media," were created for posts, users, and media, respectively.

The "null" was employed as an indicator of absence in the data files. In the "tweets" file, the fields within the columns referenced_tweet_id, mentions, URLs, hashtags, and media_keys may be null if the post does not fall into the interaction types (retweeted, quoted, or *replied_to*), lacks mentions, *URLs* or *hashtags* in the text body, or contains no media, respectively. In the "users" file, the author_location column may be null if the user has not filled in this editable text field. Additionally, in the "media" file, media_url may be null if the media type is not a photo, as the API does not provide links for videos and GIFs. The field media_view_count may be null if the media type is a photo or GIF, as they do not have a count of views.

The second phase involved searching contexts other than retweets. In this scenario, many collected posts were quotes or replies, necessitating access to the original tweet for context and

**Table 2. Tweets information.**

| Column | Definition | Type |
|---|---|---|
| id | tweet identification code | string |
| text | tweet text | string |
| created_at | date tweet was created | date |
| source | tweet source (device or connected on the platform) | string |
| lang | tweet language | string |
| conversation_id | code of identification for all replies involving the tweet | string |
| like_count | number of likes | integer |
| retweet_count | number of retweets | integer |
| quote_count | number of quotes | integer |
| reply_count | number of replies | integer |
| type | one or more types (tweeted, retweeted, quoted, replied_to) | string vector |
| referenced_tweet_id | referenced tweet identification code, in case the type is not tweet | string vector |
| mentions | users mentioned in the tweet | string vector |
| URLs | URLs in the tweet | string vector |
| hashtags | hashtags in the tweet | string vector |
| author_id | author identification code, also used to find them in the authors file | string |
| media_keys | media identification code, also used to find it in the media file | string vector |
| hashtags_count | number of items in the hashtags vector | integer |
| URLs_count | number of items in the URLs vector | integer |
| mentions_count | number of items in the mentions vector | integer |
| media_keys_count | number of items in the media_keys vector | integer |

Table representing each of the columns of the tweets files

**Table 3. Users information.**

| Column | Definiton | Type |
|---|---|---|
| account_id | código de identificação do autor | string |
| account_username | username do autor | string |
| account_created_at | data de criação da conta | data |
| account_verified | se a contaé verificada | boolean |
| account_protected | se a contaé protegida | boolean |
| account_location | texto do campo localização | string |
| account_have_profile_image | se a conta tem imagem de perfil | boolean |
| account_followers_count | quantidade de seguidores da conta | integer |
| account_following_count | quantidade de conta seguidas | integer |
| account_tweets_count | quantidade de tweets da conta | integer |

Table representing each of the columns of the users files

understanding. This process could be optimized to avoid gathering original tweets already collected. Therefore, all posts in the referenced_tweet_id fields were collected to access the original post and preserve context without. Collection was performed in blocks of 100 IDs, maintaining the same information as in the first phase and stored in the same files. This procedure meant that some older posts, predating the collection day and/or the analysis period, had to be retrieved.

Finally, a third phase focused on the contexts of tweets published in candidates' accounts. In this phase, all quotes and replies were collected, specifically those linked to the same conversation_id, representing all interactions directly or indirectly related to the initial post of the candidates' tweets. These collections served as new search contexts. The first phase was applied to each type of interaction collected. The resulting three files were stored in special directories within the folder containing the collection day information. Additionally, a fourth type of file was generated to store the simplified information of the level one ego network generated in this phase, with a file for each interaction (Table 4 describes its structure).

The JSON format used in the first phase was also adopted in the subsequent phases. It served as a central file for controlling which contexts and days had already been collected, managing completion status for each phase, and storing collection status. This facilitated resuming the process from a relatively close point in case of external events, such as machine shutdown or loss of connection, saving time and avoiding token limit issues.

## Data processing and storage

Most of the collection was conducted in a centralized manner, utilizing a dedicated server. This server remained operational twenty-four hours daily to use the available resources

**Table 4. Media information.**

| Column | Definition | Type |
|---|---|---|
| media_key | media identification code | string |
| media_type | type of media (photo, video, or GIF) | string |
| media_URL | media link | string |
| media_view_count | number of views if the media is a video | string |

Table representing each of the columns of the media files

consistently. From the initial implementation of the system, this process occurred successfully for most of the time allocated to data collection, except for a few instances where errors occurred or resource limits were encountered.

As previously noted, the volume of data remained a constant concern, influencing several decisions throughout the study. Some individual queries alone yielded approximately 5 GB of data. The challenges faced by the research team extended beyond the limitations imposed by the Twitter API; considerations such as storage, internal data sharing, and processing, including initial treatment and exploratory analysis, demanded careful planning and implementation of solutions to ensure the feasibility of the research.

With the rapid expansion of the obtained data, we recognized the imperative for enhanced formality in storage practices. Firstly, the need for backups was evident, both during the collection phase and at the moment of final storage. The data was acquired in concise blocks, each corresponding to a search context conducted on tweets from a specific day. These blocks were meticulously organized and cataloged, streamlining future data retrieval. Upon concluding the collection, the blocks were further organized into topics and restructured by merging results from collections related to each topic. This process yielded thematically enriched blocks, such as "Lula profile/ego," "Bolsonaro profile/ego," "Lula query," and "Bolsonaro query." The selection of topics was guided by their significance in the Brazilian electoral scenario, as well as their thematic and quantitative prevalence (volume of data obtained on the topics) during collection.

Each block was segmented into various files, each housing a table indexing a priority class of fields, such as User, Tweet, Media, etc. Together, these files enabled the reconstruction of entities consisting of data from one or more fields and relationships within that block. Each file was compressed in the parquet format to optimize storage efficiency, utilizing a compression algorithm optimized for CSV format tables. Finally, the file sets from each block were gathered into a *.zip file, maximizing compression. The original files' backup was retained to address any potential errors during processing phases or compression.

Sharing the collected files among team members posed a challenge due to the high volume and constant updating of the dataset, requiring centralized access, updating, and organization. Initially, configuring the collection server as a database and remote access server was proposed, but security and bureaucratic issues related to the institution the researchers are affiliated with made it necessary to find alternative solutions. Subsequently, the decision was made to store the data in the cloud. After careful consideration, Google Drive was chosen, meeting the team's organizational requirements. The Google API (Javascript and Python interface) automated certain data transfer and organization functions, facilitating efficient internal sharing. This approach allowed researchers to obtain data, or samples thereof, and update or apply corrections to the dataset using the automated server, minimizing conflicts or data losses.

Several exploratory analyses were conducted using the dataset, including those generating metrics introduced in this study. Given its aim to reflect an excerpt from a social media platform, these analyses were conducted within the context of social network analysis using the Python library pandas. Furthermore, data visualization libraries such as "Matplotlib" and "Plotly" were employed. Because the time to perform simple analyses (such as conditional sample extraction) was unsatisfactory when not optimized, it was crucial to face that this research dealt with big data.

Using non-optimized methods in the Python language, on average, simple tasks such as obtaining a random sample of user IDs took around 3 to 5 hours. Solutions were sought to process massive volumes of data to accommodate the research needs according to the planned calendar and enable more complex analyses in a reasonable timeframe.

The use of specific technologies for processing big data is considered standard nowadays. Typically, such technologies use special indexing strategies, parallel processing (often in a "cluster"), and intelligent storage strategies. Examples include Elasticsearch, Apache Hadoop, Apache Spark, and Google Big Query. Although we explored the integration of these tools, the substantial effort needed (in financial terms in some cases) led us to investigate strategies native to programming and the Python language.

As an interpreted language, Python runs slower than compiled languages, such as Java and C++. However, many of its libraries, including Pandas, serve as interfaces between Python and compiled languages. Thus, there are strategies to maximize the efficient use of computational resources by delegating the execution time to compiled and optimized functions. These functions often use modern resources present in CPUs and GPUs (such as vectorized arithmetic instructions, e.g., SIMD, Vector Processors), enabling the development of efficient algorithms from an interface language like Python, when using optimization strategies in a systematic and pragmatic way.

Much of this process can be achieved by seeking to "vectorize" operations. In this context, Vectorization is the distribution of homogeneous operations over compatible datasets (such as accumulations over a vector and matrix multiplications). Constructing situations that fit this scenario makes it possible to use explicit syntaxes relating to vectorized operations, which commonly exist considering vector structures (one-dimensional lists) or matrices.

Starting from this programming paradigm, which is part of what allows the emergence of most big data tools mentioned, it was possible to highly optimize the processing algorithms. Many were generalized to be applied to different problems or analyses related to the database. After optimization, execution times on the same machines dropped from hours to minutes, with the example of extracting a sample of user IDs taking, on average, around 40 seconds. This reduced the priority of integrating third-party systems for processing the dataset, allowing the study not to be dependent on one of these technologies.

Nevertheless, these tools can bring long-term benefits as they represent significant advancements in the field of analyzing massive amounts of data. While the optimizations carried out in the study proved to be sufficient and, in the researchers' opinion, simpler than the integration of a new system into the project, we believe that the use of a robust big data analysis system would bring advantages as new and different types of analysis were necessary, or the scale of the study grew (greater volume of data; higher data heterogeneity; more database users, in the case of a distributed database). Robust big data analysis systems optimally implement functionalities for an immense diversity of operations, coordinate strategies for highly parallelized or distributed systems, or coordinate data channels with massive flow (up to Petabytes) through networks like the web.

**The end of the academic API.**   After Elon Musk purchased Twitter, the capitalization of this process was announced, and the Academic API was discontinued, ending our data collection on July 24, 2023.

## Challenges

The Twitter API presented a significant challenge in the development of our research. The maximum number of requests allowed in a 15-minute interval rendered the collection of ego networks for the candidates' followers unfeasible. The limit of gathering only 15,000 every 15 minutes made it impossible to retrieve tweets from, for example, Bolsonaro's followers, who numbered 8.6 million at that time.

Additionally, the high volume of posts in October prompted us to alter our approach by deciding to collect reposts after obtaining the tweets. The monthly limits (60 million tweets per

month) were insufficient to cover everything, resulting in a delay in the collection process. Unfortunately, it was not possible to complete the process before the end of the Academic API. Another problem was that the retweets were truncated from the API, reducing the total number of mentions, URLs, and hashtags extracted from the texts.

Dealing with such large volumes of data was a new experience for the researchers. Designing, organizing, and manipulating a corpus of this magnitude with the available infrastructure posed a major challenge. This led to the need to re-planning and redesign tools to meet the demands, as well as refactoring the data storage, including changing the file format and keeping track of collected days to avoid unnecessary consumption of keys.

## Results

This section presents the main results of a first exploratory corpus analysis. It presents information related to the four main candidates in the 2022 Brazilian presidential elections.

### Descriptive measures of the corpus

A total of 282,135,572 posts were collected from Twitter (X), comprising tweets, retweets, quotes, and replies. Because of the functionality of the Twitter API, and a part of the used methods of collection, some of the data includes duplicate objects (due to the API, and intersecting search contexts) or items that date prior to the study's targeted period (due to the collection of referenced tweets). Among these posts, 176,254,397 were retweets, 65,680,188 were replies, 30,168,951 were tweets, 9,479,217 were quotes, and 552,819 were mixed interactions (posts with both types *quoted* and *replied_to*, shown in the graphs as *Quoted_replies*). Reposts significantly outnumbered other types, highlighting their primary role in content propagation on the social media platform.

Fig 2 depicts multiple peaks throughout the collection period, each corresponding to emblematic events during the study. On August 26, 2022, a significant spike in tweets and retweets occurred following an interview with presidential candidate Lula da Silva on the renowned Brazilian journalistic program, "Jornal Nacional" (Globo TV network). Another peak on August 29, 2022, reflected discussions among Twitter users regarding the first

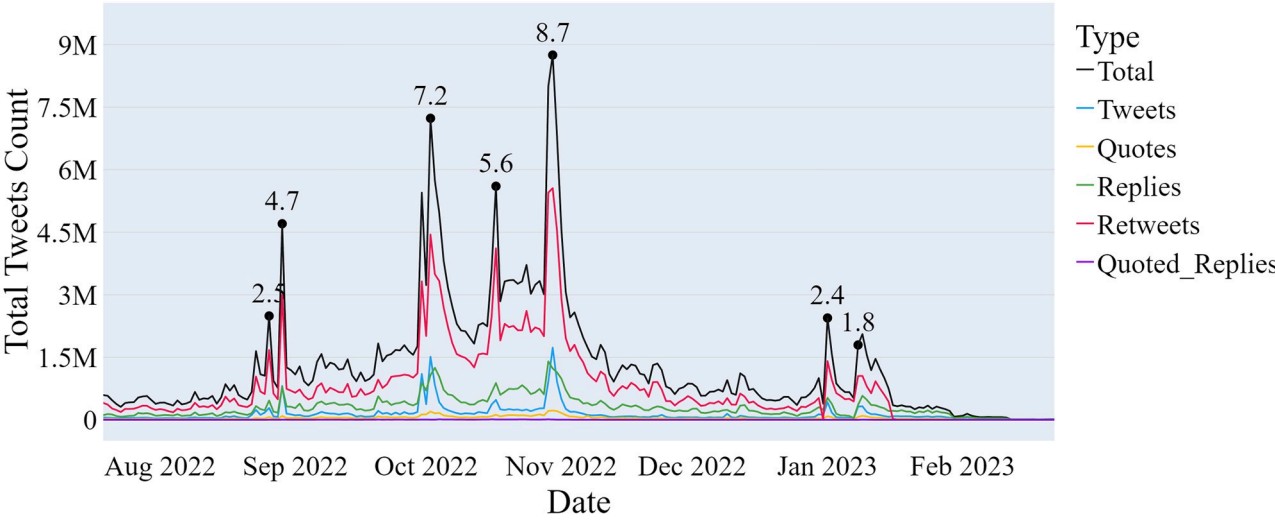

**Fig 2. Number of tweets collected over time.**

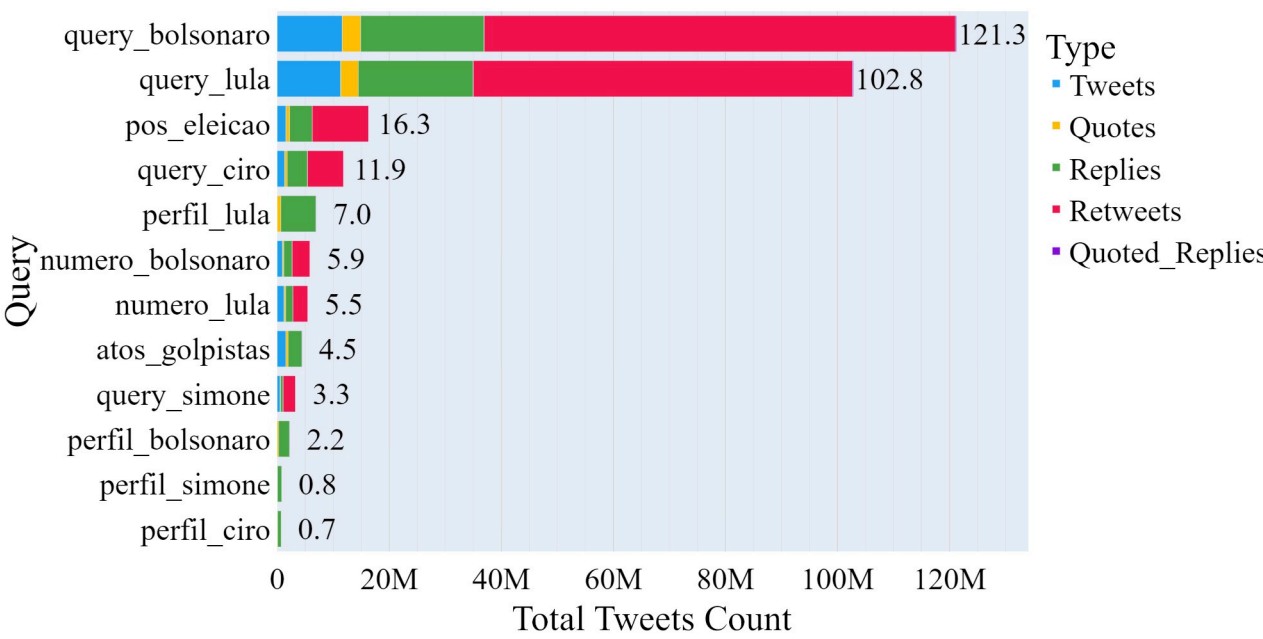

**Fig 3. Number of tweets collected per query and type.**

televised debate between presidential candidates the previous night. In October, three notable peaks were observed: the first on October 02, 2022, coinciding with the first round of elections; the second on October 17, 2022, the day after the first debate in the second round; and the third on October 30, 2022, corresponding to the second round of elections. Additionally, two peaks were noted in January: the first on January 01, 2023, marking President Lula da Silva's inauguration, and the second on January 08, 2023, when supporters of former President Jair Bolsonaro stormed the buildings of the Congress, the Supreme Court, and the Palácio do Planalto, the president's office.

Fig 3 illustrates that the search contexts for tweets and retweets related to Lula da Silva and Jair Bolsonaro dominated the corpus, significantly surpassing the number of posts concerning other presidential candidates. Additionally, the search context for Lula's profile recorded numerous interactions (Table 5), securing the sixth position among contexts with the most tweets.

A total of 303,683,456 mentions, 75,417,581 URLs, and 26,602,848 hashtags were extracted, providing information about featured accounts, fake news, and prominent hashtags. Additionally, data was collected from 280,404,916 accounts, of which only 3,076,066 were "verified accounts." Many of these were repeated due to users publishing multiple tweets in a day or

**Table 5. Information from interactions.**

| Column | Definition | Type |
|---|---|---|
| account_id | author identification code | string |
| tweet_id | tweet identification code | string |
| interaction_authors | vector with the IDs of authors who perform the interaction | string vetor |
| interaction_ids | vector with IDs of tweets that are the interaction | string vector |

Table representing each of the columns of the quote or reply files

**Table 6. Descriptive measures of the dataset.**

| Search context | Date in which 25% of the data was collected | Date in which 50% of the data was collected | Date in which 75% of the data as collected | Average data per day | Standard deviation | Total Number |
|---|---|---|---|---|---|---|
| acts related to the coup d'etat attempt | 2023/01/08 | 2023/01/12 | 2023/01/25 | 20.4 k | 55.7 k | 4.5 M |
| number bolsonaro | 2022/10/06 | 2022/10/16 | 2022/10/23 | 26.8 k | 69.8 k | 5.9 M |
| number lula | 2022/10/04 | 2022/10/17 | 2022/10/25 | 25.1 k | 72.6 k | 5.5 M |
| profile bolsonaro | 2022/09/01 | 2022/10/07 | 2022/11/16 | 10.2 k | 13.4 k | 2.2 M |
| profile ciro | 2022/08/29 | 2022/09/19 | 2022/09/29 | 3.4 k | 7.9 k | 737.2 k |
| profile lula | 2022/09/21 | 2022/10/21 | 2022/11/22 | 31.8 k | 29.4 k | 7.0 M |
| profile simone | 2022/10/03 | 2022/10/20 | 2022/11/01 | 3.8 k | 8.9 k | 835.4 k |
| post-election | 2022/11/05 | 2022/11/17 | 2022/12/10 | 74.6 k | 149.0 k | 16.3 M |
| query bolsonaro | 2022/09/25 | 2022/10/17 | 2022/11/02 | 553.7 k | 612.1 k | 121.3 M |
| query ciro | 2022/08/29 | 2022/09/20 | 2022/10/02 | 54.1 k | 101.5 k | 11.9 M |
| query lula | 2022/09/30 | 2022/10/21 | 2022/11/08 | 469.6 k | 584.8 k | 102.8 M |
| query simone | 2022/09/02 | 2022/09/30 | 2022/10/07 | 14.9 k | 39.2 k | 3.3 M |

across different days and search contexts. Repetitions allowed for the collection of temporal information, aiding in the identification of tweets published in diverse search contexts. The information collected also encompassed 30,673,896 media posts, consisting of 18,502,629 images, 11,505,363 videos, and 665,904 GIFs. Given that many of these media posts were repeated through reposts using the same media as the original post, it is feasible to download still-available media via the provided addresses. This enables the creation of a database of images and videos disseminated during the research period.

Table 6 presents descriptive measures of the corpus, with a notable high standard deviation in candidate-related data. This is a result of search contexts capturing significant spikes in platform activity compared to other days, as depicted in Fig 2. For candidates, this increase is attributed to the surge in tweets close to the first round (October 02, 2022) and second round (October 30, 2022) of the presidential elections. The dates in Table 6 indicate when 25%, 50%, and 75% of the total tweets were collected for each search context.

## Analysis of the sample

For the sample analysis, a temporal cut from the database was executed, covering the period from October 27, 2022, to November 02, 2022. This time frame encompasses the pre-election period (October 27, 28, 29, and 30) and the post-election period (October 31, November 01, and 02). Additionally, it represents the period with the highest number of tweets related to the collected hashtags.

Tables 7 and 8 showcases four randomly selected examples of tweet texts from the database within this timeframe. These tweets, diverse in content, consist of emojis, hashtags, and media. They may include news, opinions, and humorous texts, among other types of content. Additionaly, emojis were substituted by standardized literal descriptions for character encoding compatibility reasons. These descriptions are **bolded** for clarity, and have been translated to portuguese, from it's standardized english labels, in the portuguese versions of the tweets.

The word cloud, a weighted list model commonly employed for visualizing text data, serves as a vital data visualization tool. Fig 4 illustrates the word cloud generated for the studied time frame by the WordCloud [27] in Python. The cloud is formed based on the frequency of words in the text, and for this result, the NLTK [28] library in Python was utilized to remove

**Table 7. Examples of tweets texts (Portuguese).**

| | |
|---|---|
| Lula, você não foi só eleito, você fez história! [*coração, estrela*] | @Miltonneves [*mulher-levantando-mão, face-chorando-ruidozamente, seta-para-baixo*] #Urgente Traficantes assassin@m Bolsonaristas. Quem votar no Lula apoia o Crime. 22 Bolsonaro [*fogo, bandeira-brasil*] @Miltonneves https://t.co/xnQnuA2jhB |
| unica coisa que me preocupa mesmo com todo o descaso, todo o desamparo, todas as corrupções, todos os pastores, o bonoro ainda chegou no segundo turno | Com frente ampla, Lula vence eleição presidencial mais acirrada da história brasileira, derrotando a extrema direita e marcando volta ao poder das forças de centro. Bolsonaro se torna primeiro presidente a perder a reeleição. https://t.co/JfZjRL6Nx2 #ELEICOES2022 https://t.co/6sY3UpZlxq |

**Table 8. Examples of tweets texts (English).**

| | |
|---|---|
| Lula, you weren't just elected, you made history! [*heart, star*] | @Miltonneves [*woman-raising-hand, loudly-crying-face, down-arrow*] #Urgent Drug traffickers murder Bolsonaristas. Whoever votes for Lula supports Crime. 22 Bolsonaro [*fire, flag-brazil*] @Miltonneves https://t.co/xnQnuA2jhB |
| The only thing that worries me, even with all the neglect, all the helplessness, all the corruption, all the pastors, Bonoro still arrived in the second round | With a broad front, Lula wins the fiercest presidential election in Brazilian history, defeating the extreme right and marking the return to power of center forces. Bolsonaro becomes the first president to lose re-election. https://t.co/JfZjRL6Nx2 #ELEICOES2022 https://t.co/6sY3UpZlxq |

irrelevant words (stopwords). The word cloud is presented separately for the pre- and post-election periods and further categorized between candidates Lula da Silva and Jair Bolsonaro.

During the electoral period, words referring to candidate Bolsonaro frequently included terms such as "liar," "lied," "despair," "president," and "genocidal." The term "genocidal" was particularly used in reference to Bolsonaro's anti-vaccine stance during the COVID-19

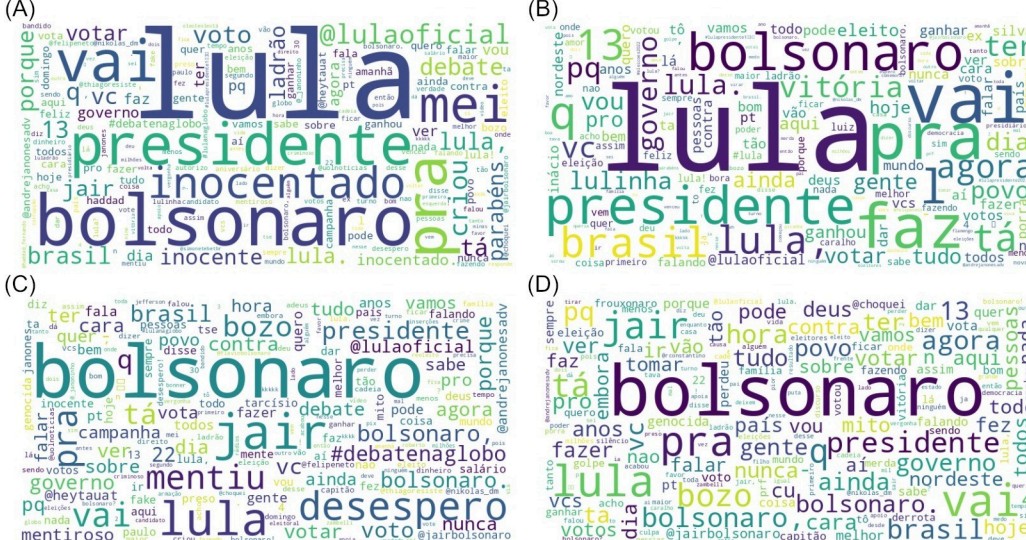

**Fig 4. Wordclouds.** #lula before the election (A), #lula after the election (B), #bolsonaro before the election (C), #bolsonaro after the election (D).

**Table 9. Table of the database statistical measures.**

|  | Lula | Bolsonaro |
|---|---|---|
| Average number of emojis per tweet | 0.43 | 0.34 |
| Percentage of tweets with media | 12.73% | 11.93% |
| Number of users who tweeted | 5.011.951 | 4.384.042 |
| Percentage of verified accounts | 2.09% | 2.27% |
| Percentage of users with active location | 56.52% | 55.99% |

pandemic. In the post-electoral period, terms reflecting Bolsonaro's defeat, such as "caiu" (fell), along with nicknames like "bozo" and "frouxonaro," became prevalent. The nickname "frouxonaro" combines "frouxo" (weak) with "Bolsonaro," serving as a metaphor for questioning his virility. Other frequently used words included "Nordeste" (Northeast), "ganhou" (won), and "votar" (vote), referring to the winning candidate, Lula.

During the electoral period, the most common words referring to candidate Lula were "president," "debate," "government," and "innocent." The term "innocent" was emphasized by Lula's supporters, highlighting the annulment of the charges that led to his arrest. A semantic and interpretative dispute arose over Lula's arrest and release, with Bolsonaro, his political opponent, using the term "unconvicted." Words expressing positive feelings, such as "happy," "good," and "truth" also appeared frequently. In the post-election period, words like "people," "won," "victory," and "Northeast" were commonly used to celebrate Lula's victory, especially in the Northeast region of Brazil.

Fig 4 presents Wordclouds separately for the pre- and post-election periods and further categorized between candidates Lula da Silva and Jair Bolsonaro.

Table 9 provides statistical measures for the given time frame. The values indicating the number of different users who tweeted are presented in absolute terms. Considering emojis in texts can be advantageous for tasks such as sentiment analysis.

Fig 5 depicts a map of Brazil based on data collected solely on October 30, 2022, showcasing the active locations of users who tweeted at least once on that date. Data processing aimed to extract as many locations as possible, given that the profile location, while not mandatory, allows users to input any characters in the text box.

## Discussion

The higher volume of tweets related to candidate Lula during October and November, with over 5 million users making publications about the then-candidate (surpassing just over 4,3 million users posting about Bolsonaro), should not necessarily be interpreted as an indication of greater support on social media platforms for Lula da Silva.

In the 2018 elections, negative propaganda from Bolsonaro's campaign targeted not only Fernando Haddad, his main opponent from the Workers' Party (PT), but also former President Lula, who was serving a sentence in Curitiba after being convicted of passive corruption, and the PT itself. Although considered a legitimate resource for providing information and stimulating debate on public policies, candidate proposals, and past achievements [30], negative propaganda took on new dimensions with the emergence of electoral campaigns on digital platforms.

The task of the Brazilian Electoral Justice in monitoring negative propaganda became more complex as multiple actors initiated attacks on the electoral process through social media platforms, adopting a non-official negative campaign strategy.

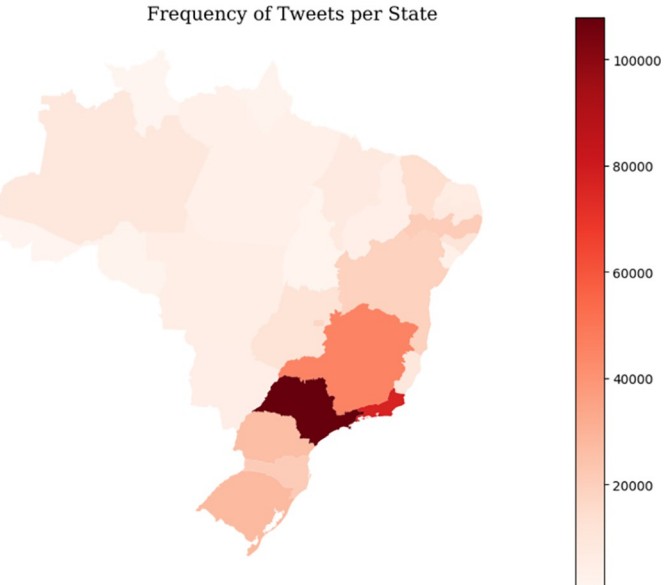

**Fig 5. Tweets about Lula and Bolsonaro on October 30, 2022.** The maps were generated using python and the python library "Geopandas" [29], with map data (contours) provided by the same library.

In this context, Jair Bolsonaro's campaign and supporter groups not directly affiliated with the campaign adopted the strategy of deconstructing the image of Lula and the PT through social media. In 2022, a consistent effort was made to link candidate Lula with corruption, and the term "thief" was prominently featured in the candidate's word cloud.

This strategy, at times, relied on controversial facts such as past judicial accusations against the candidate and, at other times, on falsehoods, such as the claim that Lula's convictions had not been annulled, which the Federal Supreme Court (STF) had done in 2021. Consequently, there was an emphasis on negative propaganda to portray negative attributes of Lula as more concerning than Bolsonaro's negative attributes, suggesting that issues like corruption and the alleged absence of moral values should be viewed with greater gravity than criticisms of Bolsonaro's handling of the Covid-19 pandemic.

Based on the exploratory analyses and the quantitative data obtained, the database holds enormous potential for analysis and knowledge acquisition. It serves as a valuable source with unique characteristics stemming from the dynamics of digital social media platforms like Twitter. These social and political data provide a snapshot of the Brazilian political landscape, particularly concerning the 2022 elections. Historically, electoral data with significant informational dimensions have been highly sought after, given their decisive potential in sociopolitical processes and power struggles, carrying profound implications for one or more nations.

The availability of data on the internet with "free" access doesn't necessarily mean that the "information" is free. Deciding on the analyses, transformations, and visualizations possible with the dataset is not trivial and might not even constitute a countable set of tasks. So far, The work represents only a small fraction of the potential explorations. Hopefully, this effort will pave the way for numerous others to contribute further to the democratization of Brazilian sociopolitical information.

## Limitations

Although the goal was to limit the amount of data preprocessing to only what was strictly necessary (leaving further refinement for future work), the possibility of conducting a thorough analysis of the presence of false positives in the final dataset was acknowledged. This would involve identifying the presence of data related to content that does not fit within the sociopolitical scope of the study or does not contribute to the overall understanding of the initially outlined research objectives. However, due to the scale of the dataset produced, there are numerous and varied methodologies that could be employed to address this issue. Most, if not all, include heuristics that may not yield completely accurate results or rely on subjective definitions, choices, and decisions regarding attempts to answer questions such as: 'Which Tweets are relevant for studies derived from this dataset?'

The potential benefit of applying such preliminary analysis to future studies based on this work is understood, yet for the reasons highlighted, it is concluded that such data preprocessing falls outside the scope of this study, allowing future work to conduct analyses within their specific contexts and leaving the decision-making power of what is relevant in the hands of those who will use the dataset produced. This conclusion is further reinforced when considering that most studies will likely use a subset of the dataset or data pertaining to a specific context (political, for example). In many cases, identifying false positives, given the broad range of potential contexts, may require considering many subtleties. These subtleties and the presence of multiple contexts justify the presence of different methodologies for selection—of which, probably the most appropriate are AI based—and which, to achieve the best results, would require a different input, training data, or fine-tuning, depending on the target context.

Finally, despite this limitation, confidence in the value of this dataset remains, given the scarcity of large datasets like this available in the academic-scientific context; and also in light of the considerations and decisions made by specialists in sociopolitical studies, which led to the preliminary observations and the construction of the queries used for data collection, as outlined in this study. We believe that future users of this dataset will be able to extract that value using existing and well-known methodologies for filtering text-based datasets chosen based on the specific objectives and/or political context that they choose to study.

## Data availability

Different strategies for making the data available were considered to ensure secure data storage and accessibility of the research results.

The primary data collected were stored on physical servers at the research group's Federal University of São Carlos headquarters. This ensures the complete dataset is in a secure location with restricted, controllable, and traceable access.

Regarding the public availability of the data, relevant legal terms were consulted—especially those applicable to the Twitter platform—along with ethical guidelines for research with Big Data and strategies used in scientific studies focused on Twitter with similar legal and availability requirements. It was found that most of the datasets from the 153 articles collected in the systematic literature review were made available on GitHub. All analyzed articles that mentioned ethical issues reported that this type of data collection is based on public data and is exempt from the Research Ethics Committee, which is the case for the present research, following national standards and resolutions. The methods of data availability for the largest corpora described in Table 1 of this article were also analyzed in detail. After thorough analysis and consideration, it was concluded that the most appropriate option would be to adopt a strategy similar to that of Chen et al. [31], who collected 1.2 billion tweets related to the 2020 US elections.

Consequently, the chosen data storage and distribution channel was GitHub [32], commonly used to organize and share software codes and configurations and allows the sharing of text files or compressed files. On GitHub, it is possible to make large volumes of data available, along with descriptions and usage licenses, without the need for payment or fees, as long as the size of each file in the repository does not exceed a certain limit in megabytes (MB).

The dataset produced consists of multiple files (in text format, compressed), organized and classified according to the queries and dates related to the data collection period and the types of objects they represent (users, tweets, or media files). Therefore, to transform the dataset into a publicly available version, the same structure was maintained without any pre-processing, only filtering the content distributed publicly.

To comply with ethical and legal principles, the published data was duly anonymized, removing all fields that could be used to identify users. This procedure prevents and avoids the inference of individual users' sensitive characteristics by third parties. Therefore, it was decided to make only the IDs of the objects in the dataset available. All information related to each object is linked to its ID and curated by X (formerly Twitter). It can be re-obtained—at the time of writing this article—through the same API used in this study. This process is commonly called "rehydration" of the data [31], and is straightforward once you have the target IDs and access to the API (which is summarized as an access key).

If researchers are interested, additional information related to accounts and users can be requested directly from X through its API in the rehydration process, which will be legally responsible for sharing it. However, this process of obtaining user account data depends on the operation of the X platform and may be subject to prices and fees imposed after its acquisition by Elon Musk. Nonetheless, the data from this research, made publicly and freely available, allows for a variety of analyses and applications. This strategy made it possible to distribute the study's data while maintaining security, integrity, and compliance with Twitter's legal terms and the privacy of individuals whose public data may be present in the dataset. Tools that facilitate the rehydration process are available on the GitHub account of the Interfaces research group, along with the dataset. A brief explanation of the process is also available, with links provided in the Data Availability statement of this article. The code developed in Python for this data collection was also made available.

## Conformity with Twitter and Twitter API's terms of use and privacy and developer policies

All steps of this research, including formulating future studies based on this research, involved attention and care to Twitter's rules for the usage of its data in the setting of the research, the resulting dataset, and its future applications. The terms agreed upon when the corresponding parties involved in the study applied for researcher access can be found in July 2022's Twitter (and Twitter API's) Terms of Use and the Developer Agreement and Policy. In this context, developer refers to anyone utilizing Twitter's services or data to collect, transform and/or redistribute Twitter content in any form.

It is possible to obtain a copy of those terms, in their totality, utilizing archiving services—from which one of the most utilized is the Internet Archive [33], with more than 896 billion web pages archived as of this writing. Copies of those terms were also saved locally for reference, and both were and will be used to support the legal conduct, and enjoyment of the fruits of this research. All the data was collected and stored following Twitter's rules regarding user protection and data security, and it will remain so. This study does not use or display any sensitive user information. It only provides analysis consistent with Twitter users' reasonable expectations of privacy, including only data posted publicly, which is also true for this study's

resulting dataset. All the data also fall under the Twitter Privacy Policy [34], which describes the uses of user-created data, on the platform, that they agree to when using the website and its services—specifically of publicly posted data.

As this research comes attached with the availability of a dataset derived from Twitter data, special attention was put towards the legality and compliance surrounding the distribution of such data, whose terms and concerns can be found primarily on Twitter's Developer Policy [35]. Those terms restrict the distribution of twitter data and twitter derived data, which are, however, eased in the context of academic research for non-commercial purposes, which will be the case for the publication of the resulting dataset. That includes the unlimited distribution of Tweets (which provides for Retweets, Comments, Quote Tweets, etc.) and User IDs. Therefore, we believe that providing access to such IDs through GitHub will be a good strategy for distributing the data available in our dataset. One example of successful usage of this strategy can be found in a 2022's study where 1,258,209,617 (more than one billion, two hundred fifty-eight thousand) tweets surrounding the 2020's American presidential elections were collected [31], and the resulting dataset, including all of the ID's retrieved in the study, were made publicly available in a GitHub Repository [36].

The entire study, including data collection and analysis, followed common safety practices to guarantee the safety and integrity of the data being collected and processed concerning the intended usage of the Twitter API. The data remained secured entirely on-site, and backups of the raw data were maintained and periodically checked to verify that the processed data matched the raw data.

Finally, it is important to note that all content provided to third parties, in any manner, shall remain subject to the same agreements and policies—and efforts will be made to ensure that all parties are informed, and agree, to these terms and policies—which is required for the compliant transfer of Twitter data. Especially, the "Twitter Controller-to-Controller (Outbound) Data Protection Addendum" specifies the transference of the legal encumbrance between controllers of the data and can serve as a basis for the applicability of all of the relevant terms and/or policies to the recipients of any provided data.

## Future research

Following the assessment of the database's potential for supporting future work, planning its infrastructure, particularly concerning data accessibility and handling, became necessary. While the results of this study already suggest interesting possibilities for use and manipulation, the potential for enhancing these aspects is evident, considering recent computational and technological advances in the context of big data.

Firstly, concerning storage, transitioning to a client-server architecture is believed to ensure better, more standardized data accessibility and increased capacity for parallel data distribution. This change would involve automating access control (currently manual) and adopting proprietary software for query solutions designed for handling large amounts of data in a distributed manner.

In this sense, the software would be installed on a "cluster" type server (computational artifacts managed in parallel) with high processing and storage capacity. This solution is commonplace for processing and distributing big data, as it effectively addresses the complexities associated with working with such data by accommodating various types of uses in the modern information environment.

Therefore, it is expected that, in the future, the database will be accessible through environments employing distributed data storage and processing software, such as Apache Hadoop (Hadoop Distributed File System (HDFS) + MapReduce) [37] and/or Apache Spark (with

HDFS or NOSQL database). Alternatively, solutions like Elasticsearch [38], a search and data analysis tool based on the REST and distributed concept, capable of handling a large volume of requests quickly and almost in real time, can be considered.

These software solutions abstract immense complexity, enabling more efficient handling of large amounts of data without the need to implement complex solutions from scratch. This facilitates the configuration of a secure, high-performance data server with extensive options for obtaining data from query languages.

Furthermore, a more appropriate architecture supports the effective retrieval and storage of media data, currently only present as links, due to the significant volume they would occupy if saved locally.

While addressing the storage and accessibility of the database is crucial for the future of this research, the generation of contextualized content holds significance for future studies on social impact topics. Analyzing and studying social and political phenomena and behaviors can contribute to the development of public policies and enhance societal understanding of representativeness, enabling the monitoring of democratic processes. Therefore, it is valuable to extract and create subsets of this database focused on specific contexts related to laws, regulations, mandates, policies, practices, traditions, values, and beliefs at the intersection of social and political life.

For instance, fake news is a critical topic associated with social media, and democracies are increasingly susceptible to its influence. Fake news has been created and distributed to deceive and manipulate, influencing opinion, altering power relations, strengthening hate groups, and fueling prejudice [39].

In this context, there is a need to construct a database specifically for fake news, serving as a benchmark to calibrate, train, and validate specific detection algorithms for Portuguese and for general positive learning algorithms.

Moreover, this new database holds potential for various scientific purposes, including the study of political and behavioral phenomena like echo chambers, accentuating political polarization, and homophily, defined as the tendency of individuals to establish relationships based on shared interests. Such phenomena are closely linked to the spread of misinformation [40].

It is worth noting that there are numerous possibilities for creating contextualized and annotated subsets, such as selecting tweets related to hate speech and toxic content, especially when associated with cyberbullying and other important social impact topics.

Finally, as tweets can be associated with images and videos, we aim to make media related to our database available. This information can prove valuable, particularly in the field of computer vision, which focuses on analyzing, interpreting, and extracting relevant information from images and videos, aiming for accurate and efficient descriptions.

## Conclusion

While Twitter is generally a "public" access platform and fits into big data standards, extracting valuable information is not trivial due to the volume, speed, and heterogeneity of data. This study concludes that acquiring informational value requires expertise not only in sociopolitical areas but also in computational and informational studies, highlighting the interdisciplinary nature of such research.

As long as we cannot democratically access the dynamics of certain important processes (in terms of democratic informational access), society will not be able to enjoy the complete democratic process.

Consequently, well-documented and scientifically studied datasets are crucial from a democratic perspective, offering society comprehensive information that illuminates historical political events.

## Acknowledgments

National Council for Scientific and Technological Development (CNPq) under grant number 420025/2023-5.

## Author Contributions

**Conceptualization:** Sylvia Iasulaitis, Bruno Cardoso Greco, Isabella Vicari.

**Data curation:** Alan Demétrius Baria Valejo, Bruno Cardoso Greco, Vinicius Gonçalves Perillo, Guilherme Henrique Messias.

**Formal analysis:** Sylvia Iasulaitis, Vinicius Gonçalves Perillo, Guilherme Henrique Messias, Isabella Vicari.

**Funding acquisition:** Sylvia Iasulaitis.

**Investigation:** Alan Demétrius Baria Valejo, Isabella Vicari.

**Methodology:** Sylvia Iasulaitis, Alan Demétrius Baria Valejo, Bruno Cardoso Greco, Vinicius Gonçalves Perillo.

**Project administration:** Sylvia Iasulaitis.

**Resources:** Sylvia Iasulaitis.

**Software:** Alan Demétrius Baria Valejo, Bruno Cardoso Greco, Vinicius Gonçalves Perillo.

**Supervision:** Sylvia Iasulaitis.

**Validation:** Alan Demétrius Baria Valejo, Bruno Cardoso Greco.

**Visualization:** Vinicius Gonçalves Perillo, Guilherme Henrique Messias.

**Writing – original draft:** Alan Demétrius Baria Valejo, Bruno Cardoso Greco, Vinicius Gonçalves Perillo, Isabella Vicari.

**Writing – review & editing:** Sylvia Iasulaitis.

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
