## [Decision Letter · Decision Letter 0]

21 Jun 2024

PONE-D-24-03586The Interfaces Twitter Elections Dataset: construction process and characteristics of Big Social Data during the 2022 presidential elections in BrazilPLOS ONE

Dear Dr. Iasulaitis,

Thank you for submitting your manuscript to PLOS ONE. After careful consideration, we feel that it has merit but does not fully meet PLOS ONE’s publication criteria as it currently stands. Therefore, we invite you to submit a revised version of the manuscript that addresses the points raised during the review process.

 The manuscript presents a substantial data collection campaign focusing on Twitter activity during the Brazilian presidential elections of 2022. The main contribution is the dataset, which is deemed to be of significant value for computational social science and political science research. The data collection is unique due to its focused scope, comprehensive coverage, and the volume of data generated. However, the reviewers have identified several major areas that need to be addressed before publication. 

Major Points for Revision

Data Sharing and Volume:

Issue: More details are required about the nature, volume, and mode of data sharing, including any restrictions.

Action: Provide comprehensive information on how the data will be shared, including specifics on the nature and volume of the data, and how researchers can access it. Address any restrictions that may apply, especially concerning the full text of tweets and user information in accordance with Twitter’s terms of service (ToS).

Compliance with Twitter's Terms of Service:

Issue: Concern about the legality of sharing full tweets and user information according to Twitter’s ToS.

Action: Outline the restrictions from Twitter’s ToS and explain how these will be navigated. Ensure that all shared data complies with these terms and provide a clear plan for how this compliance will be maintained.

Ethics Statement:

Issue: Lack of an ethics statement addressing the collection and aggregation of potentially sensitive data.

Action: Include a detailed ethics statement. Explain why ethical considerations do or do not apply, and whether a human subject review was conducted or exempted by the authors’ institution.

Data Storage and Repository:

Issue: Details on what will be available on GitHub, including file formats and data preprocessing, are lacking. GitHub storage limitations are also a concern.

Action: Clarify the types of files that will be available on GitHub, their formats, and any preprocessing done. Consider using scientific repositories like Dryad or Zenodo for long-term archival and provide detailed information on how the data will be stored and accessed.

Data Validation:

Issue: Unclear validation of the relevance of content downloaded when querying for candidate nicknames.

Action: Describe the methods used to validate that the downloaded content is relevant, especially for generic candidate nicknames. Provide specific examples or methodologies used to ensure data accuracy.

Minor Comments for Revision

Clarify Terminology:

Page 2, Line 41: Replace "data mining" with a more precise term as it usually does not refer to data gathering.

Page 2, Lines 47-49: Clarify the connection between AI and "geographically dispersed" datasets.

Literature Review:

Explicitly mention if the datasets reviewed share tweets or have usage restrictions.

Correct Errors:

Page 11, Line 361: Correct the Twitter handle for Ciro Gomes.

Page 14, Line 476: Indicate if there was an optimization to check if the reference ID had been encountered before.

Enhance Interpretation:

Table 7, Figure 4: Provide additional interpretation, especially for non-Portuguese readers.

By addressing these major and minor issues, the manuscript will be significantly strengthened, ensuring it meets publication standards and provides clear, valuable information to the research community.

We look forward to receiving your revised manuscript.

Kind regards,

Gábor Vattay, PhD, DSc

Academic Editor

PLOS ONE

Journal Requirements:

4. Thank you for stating the following financial disclosure: "This work is supported by FAPESP under grant number 2022/03090-0 under the coordination of the Prof.a Dr.a Sylvia Iasulaitis and BCO number 2023/03704-0 and number 2023/17214-5. This research was funded by the Coordenação de Aperfeiçoamento de Pessoal de Nível Superior - Brasil (CAPES) - Finance Code 001."

5. Thank you for stating the following in the Acknowledgments Section of your manuscript: "This work is supported by FAPESP under grant number 2022/03090-0 under the coordination of the Prof.a Dr.a Sylvia Iasulaitis and BCO number 2023/03704-0 and number 2023/17214-5. This research was funded by the Coordenação de Aperfeiçoamento de Pessoal de Nível Superior - Brasil (CAPES) - Finance Code 001."

Please remove any funding-related text from the manuscript and let us know how you would like to update your Funding Statement. Currently, your Funding Statement reads as follows: "This work is supported by FAPESP under grant number 2022/03090-0 under the coordination of the Prof.a Dr.a Sylvia Iasulaitis and BCO number 2023/03704-0 and number 2023/17214-5. This research was funded by the Coordenação de Aperfeiçoamento de Pessoal de Nível Superior - Brasil (CAPES) - Finance Code 001."

7. Please upload a copy of Supporting Information Figure/Table/etc. Tables 1,2,3,4,5,6,7,8 and Figures 1,2,3,4,5, and 6 which you refer to in your text on pages 24 and 25.

Reviewers' comments:

Reviewer's Responses to Questions

**Comments to the Author**

1. Is the manuscript technically sound, and do the data support the conclusions?

Reviewer #1: Yes

Reviewer #2: Partly

2. Has the statistical analysis been performed appropriately and rigorously? 

Reviewer #1: N/A

Reviewer #2: N/A

3. Have the authors made all data underlying the findings in their manuscript fully available?

Reviewer #1: Yes

Reviewer #2: No

4. Is the manuscript presented in an intelligible fashion and written in standard English?

Reviewer #1: Yes

Reviewer #2: Yes

5. Review Comments to the Author

Reviewer #1: This paper describes a very important and substantial data collection campaign focusing on Twitter and the Brazilian presidential elections of 2022. The authors go in detail to present the context, the challenges encountered and some solutions, along with some statistical overview of the dataset.

I believe that the main contribution in the paper is the dataset itself, that will be of immense value to computational social science and political science researchers in the future. The data collection campaign described in this article is unique in its focused scope, comprehensive topical and temporal coverage and the amount of data generated. Based on this, I find this work clearly worthy of publiction (subject to some clarifying questions, see below). However, I'm unsure of PLOS ONE's policy about publishing primarily dataset papers -- my recommendation is based on the assumption that this is indeed in scope, but I leave the decision on this (or whether to e.g. transfer to an adequate sibling journal) to the editors.

Having said the above, I have the following questions that I believe should be addressed before publication:

1. The main contribution is the presentation of the dataset itself, including that it will be available to other researchers upon publication (as claimed in the "data availability statement"). I find this of immense value, however, given the issues of sharing / redistributing this volume of data, I believe that more details should be given about the nature, volume and mode of data sharing, along with any restrictions.

2. Specifically, I'm concerned whether sharing the full text of tweets, along with information about users (Tables 2-5) is in accordance with Twitter's terms of use (ToS) and policies. I know that in the past, public sharing of data was hampered by restrictive license on tweet content. I believe that the authors should outline (either in the article, or as declarations to the editors) what restrictions from the ToS might apply, and how these can be overcome.

3. I'm surprised that the authors provide no ethics statement for this article. While I understand that the dataset includes the collection of data that was shared publicly by its creators, collection and aggregation of sensitive data (such as political opinions) can still raise ethical issues. I believe that at least a detailed explanation of why such considerations do not apply (along with details whether a "human subject" review took place at the authors institution, or whether the study was exempted) would be necessary.

4. The authors state that "All files will be made available on GitHub". Please outline what this entails in more detail (i.e. file format -- whether it is the raw download results, or after preprocessing, etc.), the size of the dataset, and any anticipated issues related to storage (as far as I know, regular Github repositories have storage limitations). Given the important nature of the dataset, I would also recommend the authors to consider depositing in scientific repositories (e.g. Dryad, Zenodo or similar) that pledge long-term archival as well.

5. Regarding data collection, a very important question is how the authors validated that the content they downloaded, especially when querying for the "nicknames" of candidates (which seem to be quite generic) are indeed relevant?

I have some further, minor comments:

-- Page 2, line 41: "data mining" is a bit unclear, since it is usually not about "gathering" data

-- Page 2, lines 47-49: it is unclear to me what is the connection of AI to "geographically dispersed" datasets

-- literature review section: it would be good to mention explicitly if the datasets share tweets themselves, or any restrictions on use

-- Page 11, line 361: Twitter handle for Ciro Gomes is likely wrong

-- Page 14, line 476: was there an optimization to check if the references ID was encountered before?

-- Table 7, Fig. 4: it would be good to show some more interpretation (especially for readers who don't understand portoguese)

Reviewer #2: The dataset has interest due to the topic, as the authors rightly argue. The pipeline is described in an understandable way, yet innovation in this field is particularly difficult. The authors could publish the dataset with accompanying documentation in a public repository, or, for eventual publication, they could deepen the analysis in some methodologically or theoretically meaningful direction. I wish the authors the best with this interesting dataset from which I am sure interesting results will emerge.

6. PLOS authors have the option to publish the peer review history of their article (what does this mean?). If published, this will include your full peer review and any attached files.

Reviewer #1: No

Reviewer #2: No

---

## [Author Response · Author response to Decision Letter 0]

29 Jul 2024

1. Major Points for Revision

1.1 Data Sharing and Volume:

Issue: More details are required about the nature, volume, and mode of data sharing, including any restrictions.

Action: Provide comprehensive information on how the data will be shared, including specifics on the nature and volume of the data, and how researchers can access it. Address any restrictions that may apply, especially concerning the full text of tweets and user information in accordance with Twitter’s terms of service (ToS).

Solution: We have included the “Data Availability” section in the article.

1.2 Compliance with Twitter’s Terms of Service:

Issue: Concern about the legality of sharing full tweets and user information according to Twitter’s ToS.

Action: Outline the restrictions from Twitter’s ToS and explain how these will be navigated. Ensure that all shared data complies with these terms and provide a clear plan for how this compliance will be maintained.

Solution: We included the section “Conformity with Twitter and Twitter API’s Terms of Use and Privacy and Developer Policies” with all the details.

1.3 Ethics Statement:

Issue: Lack of an ethics statement addressing the collection and aggregation of potentially sensitive data.

Action: Include a detailed ethics statement. Explain why ethical considerations do or do not apply, and whether a human subject review was conducted or exempted by the authors’ institution.

Response:

The Ethics Committee of the authors’ institution follows the Brazilian standards applied to scientific research, which are provided in its resolutions available at Reso510.pdf (gov.br) (link) e Frequently Asked Question — (ufscar.br) (link) that:

Some types of research use methodologies characteristic of the Human and Social Sciences and are exempt from submission to the CEP/Conep System, as provided for in the National Council of Health (CNS) Resolution 510/2016 in its Article 1:

Paragraph one: The following will not be recorded or evaluated by the CEP/CONEP system:

I – public opinion survey with unidentified participants;

II – research that uses publicly accessible information under the terms of Law 12527 of November 18, 2011;

III – research that uses information of public domain;

IV – census research;

V – research using databases with aggregated information, where it is impossible to obtain individual identification.

This research falls within the terms of this resolution and its submission for consideration by the Research Ethics Committee is not required, since:

a. the disclosure of the dataset will not allow the identification of participants, as only the ID will be made available and not individual personal information;

b. the data obtained from Twitter was, at the time of its collection, available for access by researchers, who carried out the collection in accordance with Twitter's ToS;

c. the research uses tweets that were made publicly and can be consulted, used, and reproduced without copyright or intellectual property restrictions, so authorization is not necessary, in accordance with the provisions of the Twitter ToS (detailed in the section “Conformity with Twitter and Twitter API's Terms of Use, and privacy and developer policies” inserted in the article) and the Brazilian legal system on public domain works.

The primary data collected are stored on physical servers on the premises of the research group’s home institution, the Federal University of São Carlos. Therefore, the complete dataset is in a secure location, access to which is restricted, controllable, and traceable.

The 153 articles consulted to carry out the literature review, when they mention ethical issues, inform that Twitter data is public and, therefore, exempt from approval by the Research Ethics Committee.

The 13 articles with the largest corpus available in Table 1 of our article also did not go through an Ethics Committee, according to the spreadsheet prepared with this information:

https://docs.google.com/spreadsheets/d/12Ozw7_fhy1RnH-9kI5eczorQVtxCw1fnHMSL5DJHwks/edit?gid=197341791#gid=197341791

Article Title Corpus Size Is there any information about data availability? Where is the data located? Is there any information on ethical issues?

1.4 Data Storage and Repository:

Issue: Details on what will be available on GitHub, including file formats and data preprocessing, are lacking. GitHub storage limitations are also a concern.

Action: Clarify the types of files that will be available on GitHub, their formats, and any preprocessing done. Consider using scientific repositories like Dryad or Zenodo for long-term archival and provide detailed information on how the data will be stored and accessed.

Solution: The data is readily available on the GitHub account of the Interfaces research group for the purpose of evaluation by the PLOS ONE editor and reviewers, together with the requested description. The access link is https://github.com/Interfaces-UFSCAR/ITED-Br?tab=readme-ov-file

1.5 Data Validation:

Issue: Unclear validation of the relevance of content downloaded when querying for candidate nicknames.

Action: Describe the methods used to validate that the downloaded content is relevant, especially for generic candidate nicknames. Provide specific examples or methodologies used to ensure data accuracy.

Solution: We have inserted specific examples in the article.

2. Minor Comments for Revision

2.1 Clarify Terminology:

Page 2, Line 41: Replace “data mining” with a more precise term as it usually does not refer to data gathering.

Solution: The request was met, with the term being changed to “data scraping.”

Page 2, Lines 47-49: Clarify the connection between AI and “geographically dispersed” datasets.

Solution: This has been removed.

2.2 Literature Review:

Explicitly mention if the datasets reviewed share tweets or have usage restrictions.

Response: In the “Supporting information” section, all links to the available datasets were inserted, and hyperlinks made in the titles of the articles in Table 1.

A spreadsheet was also created with information related to the 13 largest corpus cited in Table 1 of the article in the systematic review for internal use by the editor and reviewers:

https://docs.google.com/spreadsheets/d/12Ozw7_fhy1RnH-9kI5eczorQVtxCw1fnHMSL5DJHwks/edit?gid=197341791#gid=197341791

2.3 Correct Errors:

a. Page 11, Line 361: Correct the Twitter handle for Ciro Gomes.

Solution: We made the necessary changes.

b. Page 14, Line 476: Indicate if there was an optimization to check if the reference ID had been encountered before.

Solution: We made the necessary changes.

2.4 Enhance Interpretation:

Table 7, Figure 4: Provide additional interpretation, especially for non-Portuguese readers.

Solution: A new table (8) was created with the translation of the content of table 7. In relation to figure 4, an additional explanation was made in the article.

2.5 Please include the following items when submitting your revised manuscript:

A rebuttal letter that responds to each point raised by the academic editor and reviewer(s). You should upload this letter as a separate file labeled ‘Response to Reviewers’.

Solution: We made the necessary changes.

2.6 A marked-up copy of your manuscript that highlights changes made to the original version. You should upload this as a separate file labeled ‘Revised Manuscript with Track Changes’.

Solution: We made the necessary changes.

2.7 An unmarked version of your revised paper without tracked changes. You should upload this as a separate file labeled ‘Manuscript’.

Solution: We made the necessary changes.

2.8 We request changes to our financial disclosure in accordance with the following:

This work was funded by FAPESP under grant number 2022/03090-0 under the coordination of Prof. Dr. Sylvia Iasulaitis and BCO number 2023/03704-0 and number 2023/17214-5. This research was supported by the Coordination for the Improvement of Higher Education Personnel - Brazil (CAPES) - Finance Code 001, through the granting of a master’s scholarship. (Cover Letter)

2.9 Guidelines for resubmitting your figure files are available below the reviewer comments at the end of this letter.

While revising your submission, please upload your figure files to the Preflight Analysis and Conversion Engine (PACE) digital diagnostic tool, https://pacev2.apexcovantage.com/

PACE helps ensure that figures meet PLOS requirements. To use PACE, you must first register as a user. Registration is free. Then, login and navigate to the UPLOAD tab, where you will find detailed instructions on how to use the tool. If you encounter any issues or have any questions when using PACE, please email PLOS at figures@plos.org. Please note that Supporting Information files do not need this step.

Solution: We made the necessary changes.

3. Journal Requirements:

3.1 When submitting your revision, we need you to address these additional requirements.

Please ensure that your manuscript meets PLOS ONE’s style requirements, including those for file naming. The PLOS ONE style templates can be found at 

Solution: We made the necessary changes.

3.2 Please note that PLOS ONE has specific guidelines on code sharing for submissions in which author-generated code underpins the findings in the manuscript. In these cases, all author-generated code must be made available without restrictions upon publication of the work. Please review our guidelines at https://journals.plos.org/plosone/s/materials-and-software-sharing#loc-sharing-code and ensure that your code is shared in a way that follows best practice and facilitates reproducibility and reuse.

Solution: The code was shared on GitHub, at the link https://github.com/Interfaces-UFSCAR/Codigo-Coleta-PLOS-ONE

3.3 We note that the grant information you provided in the ‘Funding Information’ and ‘Financial Disclosure’ sections do not match. When you resubmit, please ensure that you provide the correct grant numbers for the awards you received for your study in the ‘Funding Information’ section.

Solution: We made the necessary changes.

3.4 Thank you for stating the following financial disclosure: “This work is supported by FAPESP under grant number 2022/03090-0 under the coordination of the Prof.a Dr.a Sylvia Iasulaitis and BCO number 2023/03704-0 and number 2023/17214-5. This research was funded by the Coordenação de Aperfeiçoamento de Pessoal de Nível Superior - Brasil (CAPES) - Finance Code 001.”

Please state what role the funders took in the study. If the funders had no role, please state: “The funders had no role in study design, data collection and analysis, decision to publish, or preparation of the manuscript.”

Response: “The funders had no involvement in the study design, data collection and analysis, decision to publish, or preparation of the manuscript”. (Cover Letter)

3.5 Thank you for stating the following in the Acknowledgments Section of your manuscript: “This work is supported by FAPESP under grant number 2022/03090-0 under the coordination of the Prof.a Dr.a Sylvia Iasulaitis and BCO number 2023/03704-0 and number 2023/17214-5. This research was funded by the Coordenação de Aperfeiçoamento de Pessoal de Nível Superior - Brasil (CAPES) - Finance Code 001.”

Please remove any funding-related text from the manuscript and let us know how you would like to update your Funding Statement. Currently, your Funding Statement reads as follows: “This work is supported by FAPESP under grant number 2022/03090-0 under the coordination of the Prof.a Dr.a Sylvia Iasulaitis and BCO number 2023/03704-0 and number 2023/17214-5. This research was funded by the Coordenação de Aperfeiçoamento de Pessoal de Nível Superior - Brasil (CAPES) - Finance Code 001.”

Solution: We made the necessary changes.

This work was funded by FAPESP under grant number 2022/03090-0 under the coordination of Prof. Dr. Sylvia Iasulaitis and BCO number 2023/03704-0 and number 2023/17214-5. This research was supported by the Coordination for the Improvement of Higher Education Personnel - Brazil (CAPES) - Finance Code 001, through the granting of a master’s scholarship. (Cover Letter)

3.6 When completing the data availability statement of the submission form, you indicated that you will make your data available on acceptance. We strongly recommend all authors decide on a data sharing plan before acceptance, as the process can be lengthy and hold up publication timelines. Please note that, though access restrictions are acceptable now, your entire data will need to be made freely accessible if your manuscript is accepted for publication. This policy applies to all data except where public deposition would breach compliance with the protocol approved by your research ethics board. If you are unable to adhere to our open data policy, please kindly revise your statement to explain your reasoning and we will seek the editor’s input on an exemption. Please be assured that, once you have provided your new statement, the assessment of your exemption will not hold up the peer review process.

Solution: The data has already been made available on the GitHub account of the Interfaces research group for the purpose of evaluation by the editor and reviewers of PLOS ONE. The access link is https://github.com/Interfaces-UFSCAR/ITED-Br

3.7 Please upload a copy of Supporting Information Figure/Table/etc. Tables 1,2,3,4,5,6,7,8 and Figures 1,2,3,4,5, and 6 which you refer to in your text on pages 24 and 25.

Solution: We made the necessary changes.

4. Reviewers’ comments:

4.1 Specifically, I’m concerned whether sharing the full text of tweets, along with information about users (Tables 2-5) is in accordance with Twitter’s terms of use (ToS) and policies. I know that in the past, public sharing of data was hampered by restrictive license on tweet content. I believe that the authors should outline (either in the article, or as declarations to the editors) what restrictions from the ToS might apply, and how these can be overcome.

Solution: We specified the response in the article in the section “Data Availability.” Only tweet ID information will be made available, information from tables 2 to 5 will not be shared publicly.

4.2 The authors state that “All files will be made available on GitHub”. Please outline what this entails in more detail (i.e. file format -- whether it is the raw download results, or after preprocessing, etc.), the size of the dataset, and any anticipated issues related to storage (as far as I know, regular Github repositories have storage limitations). 

Solution: The data has already been made available on the GitHub account of the Interfaces research group for the purpose of evaluation by the PLOS ONE editor and reviewers, together with the requested description. The access link is https://github.com/Interfaces-UFSCAR/ITED-Br

4.3 The authors could publish the dataset with accompanying documentation in a public repository.

Solution: We made the necessary changes. The access link is: https://github.com/Interfaces-UFSCAR/ITED-Br

---

## [Decision Letter · Decision Letter 1]

7 Oct 2024

PONE-D-24-03586R1The Interfaces Twitter Elections Dataset: Construction process and characteristics of Big Social Data during the 2022 presidential elections in BrazilPLOS ONE

Dear Dr. Iasulaitis,

Thank you for submitting your manuscript to PLOS ONE. After careful consideration, we feel that it has merit but does not fully meet PLOS ONE’s publication criteria as it currently stands. Therefore, we invite you to submit a revised version of the manuscript that addresses the points raised during the review process.

We thank you for responding to the reviewers' earlier concerns. However, we request some additional detail and clarification in several parts of the submission. Specifically, please provide:

- A clearer response to Reviewer 1's comments on the use of broad terms and false positives- if the authors aren't able to do some validation as suggested, then this may be discussed as a limitation in the Discussion

- Additional supporting references, particularly in the Discussion. Specifically, this section contains statements that are not directly supported by the results or other citations. For example, the authors say "Negative propaganda against opponents, a resource commonly employed in electoral disputes, has played a prominent role in Jair Bolsonaro’s campaigns since the 2018 Brazilian presidential election." This would need a supporting reference or removal.

- In addition to the above point, the authors state "In such scenarios, the potential for negative propaganda through disinformation arises, as the connection between official coalitions or party federations (from 2022) and unofficial groups responsible for disseminating campaign content cannot be easily proven [32], making it challenging to penalize those involved." This seems to be saying that they can't confirm the campaign members were directly involved. As such, we request that the authors simply report objectively what happened here, rather than framing it speculatively as is.

- Clarification on what the date range of published tweets that were included in the study are

- Finally, Fig 5 appears to be copyrighted, and we require clarification on the appropriate permissions for this image

We look forward to receiving your revised manuscript.

Kind regards,

Avanti Dey, PhD

Staff Editor

PLOS ONE
---

## [Author Response · Author response to Decision Letter 1]

25 Oct 2024

RESPONSE TO REVIEWERS

Dear Academic Editor

M.D Gábor Vattay, PhD, DSc

PLOS ONE 

We would like to express our gratitude to PLOS ONE journal for the opportunity to review the article “The Interfaces Twitter Elections Dataset: construction process and characteristics of Big Social Data during the 2022 presidential elections in Brazil” and for the valuable comments that have helped us to improve our work. 

The authors carefully considered each of the reviewers’ and editor’s comments and did their best to address each in a revised version of the article.

After this careful review, we hope the manuscript meets the high standards and requirements of PLOS ONE. Below, we describe the changes made to the text considering each recommendation.

Finally, we clarify that all changes are highlighted in blue in a revised version of the manuscript.

Yours sincerely, 

The authors 

1. A clearer response to Reviewer 1's comments on the use of broad terms and false positives- if the authors aren't able to do some validation as suggested, then this may be discussed as a limitation in the Discussion.

Solution: We have included the “Limitations” section in the article.

Although the goal was to limit the amount of data preprocessing to only what was strictly necessary (leaving further refinement for future work), the possibility of conducting a thorough analysis of the presence of false positives in the final dataset was acknowledged. This involves identifying data related to content that does not fit within the sociopolitical scope of the study or does not contribute to the understanding of the initially outlined research objectives. However, due to the scale of the dataset produced, there are numerous and varied methodologies that could be employed to address this issue. Most, if not all, include heuristics that may not yield completely accurate results or rely on subjective definitions, choices, and decisions regarding the attempts to answer questions such as: 'Which Tweets are relevant for studies derived from this dataset?'

The potential benefit of applying such preliminary analysis to future studies based on this work is understood, yet for the reasons highlighted, it is concluded that such data preprocessing falls outside the scope of this study, allowing future work to conduct analyses within their specific contexts and leaving the decision-making power of what is relevant in the hands of those who will use the dataset produced. This conclusion is further reinforced when considering that most studies will likely use a subset of the dataset or data pertaining to a specific context (political, for example). In many cases, identifying false positives, given the broad range of potential contexts, may require considering many subtleties. These subtleties and the presence of multiple contexts justify the presence of different methodologies for selection - of which, probably the most appropriate are AI based - and which, to achieve the best results, would require a different input, training data, or fine-tuning, depending on the target context.

Finally, despite this limitation, confidence in the value of this dataset remains, given the scarcity of large datasets like this available in the academic-scientific context; and also in light of the considerations and decisions made by specialists in sociopolitical studies, which led to the preliminary observations and the construction of the queries used for data collection, as outlined in this study. We believe that future users of this dataset will be able to extract that value using existing and well-known methodologies for filtering text-based datasets chosen based on the specific objectives and/or political context that they choose to study.

2. Additional supporting references, particularly in the Discussion. Specifically, this section contains statements that are not directly supported by the results or other citations. For example, the authors say "Negative propaganda against opponents, a resource commonly employed in electoral disputes, has played a prominent role in Jair Bolsonaro’s campaigns since the 2018 Brazilian presidential election." This would need a supporting reference or removal.

Solution: This has been removed.

3. In addition to the above point, the authors state "In such scenarios, the potential for negative propaganda through disinformation arises, as the connection between official coalitions or party federations (from 2022) and unofficial groups responsible for disseminating campaign content cannot be easily proven [32], making it challenging to penalize those involved." This seems to be saying that they can't confirm the campaign members were directly involved. As such, we request that the authors simply report objectively what happened here, rather than framing it speculatively as is.

Solution: This has been removed.

4. Clarification on what the date range of published tweets that were included in the study are.

Response:

The period of analysis of the data presented in the article can be found on page 20 in the section "Analysis of the sample," namely:

The sample analysis was based on a temporal cut of the database, covering the period from October 27, 2022, to November 02, 2022. This time frame encompasses the pre-election days (October 27, 28, 29, and 30) and the post-election days (October 31, November 01, and 02).

5. Fig 5 appears to be copyrighted, and we require clarification on the appropriate permissions for this image.

Solution: Inserted the source in the image description in the original article: Source: instagram.com/tiago_asmar and facebook.com/cirogomesoficial

Response: Considering Brazilian legislation, according to Law No. 9,610 of February 19, 1998, Chapter IV, Article 46, the reproduction of certain copyrighted material (of which classification the subject images fall under) in scientific works does not constitute a violation of copyright (available at http://www.planalto.gov.br/Ccivil_03/leis/L9610.htm). Regarding U.S. legislation, the Fair Use policy available on the X platform itself (https://help.x.com/en/rules-and-policies/fair-use-policy) considers the image as "Transformative for educational and non-commercial use."

In addition to what applies under the Fair Use policy, summarizing what is relevant regarding image copyright on X (formerly Twitter) – content (object: Tweet, User, Media, etc.) extracted from the platform via the API, under terms applicable to the researchers at the time of data collection, may be distributed in its entirety at a volume of 50,000 per day per consumer. In this case (pertaining to the to-be-published article), two objects (2 images) are shared with each article reader. The remainder of the dataset, which will be shared via GitHub, is not included in this count, as it contains only the IDs of each object, which, according to the same terms, may be shared without restrictions in terms of quantity or volume.

Regarding creative authorship, efforts were made to identify the authorship of the two images that make up Figure 5 (A and B) through backtracking with search platforms. Two social media profiles were found (Instagram for Figure 5A; Facebook for Figure 5B), with high confidence that these are the original authors of the images used. This is corroborated by the fact that both images, in at least one instance each, are credited to the aforementioned authors by well-regarded news outlets in Brazil.

For the Facebook and Instagram platforms (where the images were originally identified), Meta’s Fair Use rules apply, similar to X (available at https://www.meta.com/help/policies/meta-policies/meta-copyright/).

To emphasize, it is believed, again with high confidence, that under U.S. law, the intended use of these images can be classified as Fair Use, as they are: 1- used for scientific purposes; 2- of low creative value (both are simple edits of Brazilian political figures); and 3- used in a way that will not impact their monetization (if they are monetized).

Furthermore, the original profiles will be credited under each image for ethical purposes, and, at the end of the text, a short paragraph was added clarifying the situation regarding the copyright of the images, stating: "The images from Figure 5 (A and B) are included under Fair Use for scholarly purposes. All rights to the original images remain with their respective copyright holders."

---

## [Decision Letter · Decision Letter 2]

15 Dec 2024

The Interfaces Twitter Elections Dataset: Construction process and characteristics of Big Social Data during the 2022 presidential elections in Brazil

PONE-D-24-03586R2

Dear Dr. Iasulaitis,

We’re pleased to inform you that your manuscript has been judged scientifically suitable for publication and will be formally accepted for publication once it meets all outstanding technical requirements.

Kind regards,

Nazim Taskin

Academic Editor

PLOS ONE

Additional Editor Comments (optional):

Dear authors,

I have gone through your manuscript and comments from the reviewers. I believe that you have done good job with the responses and improved your manuscript significantly. As it is on an interesting topic, developed rigorously and contributes to the field, I suggest your manuscript is published.

Good luck with your research.

Regards,

Nazim Taskin

Reviewers' comments:

Reviewer's Responses to Questions

**Comments to the Author**

1. If the authors have adequately addressed your comments raised in a previous round of review and you feel that this manuscript is now acceptable for publication, you may indicate that here to bypass the “Comments to the Author” section, enter your conflict of interest statement in the “Confidential to Editor” section, and submit your "Accept" recommendation.

Reviewer #1: All comments have been addressed

Reviewer #3: All comments have been addressed

2. Is the manuscript technically sound, and do the data support the conclusions?

Reviewer #1: (No Response)

Reviewer #3: Yes

3. Has the statistical analysis been performed appropriately and rigorously? 

Reviewer #1: (No Response)

Reviewer #3: Yes

4. Have the authors made all data underlying the findings in their manuscript fully available?

Reviewer #1: (No Response)

Reviewer #3: Yes

5. Is the manuscript presented in an intelligible fashion and written in standard English?

Reviewer #1: (No Response)

Reviewer #3: Yes

6. Review Comments to the Author

Reviewer #1: (No Response)

Reviewer #3: The manuscript presents solid, well-explained, and ethically conducted research. The dataset is a useful asset for researchers.

7. PLOS authors have the option to publish the peer review history of their article (what does this mean?). If published, this will include your full peer review and any attached files.

Reviewer #1: No

Reviewer #3: **Yes: **Yunus Emre Bulut

---

## [Editor Report · Acceptance letter]

2 Jan 2025

PONE-D-24-03586R2 

PLOS ONE

Dear Dr. Iasulaitis, 

I'm pleased to inform you that your manuscript has been deemed suitable for publication in PLOS ONE. Congratulations! Your manuscript is now being handed over to our production team.

Kind regards, 

on behalf of

Dr. Nazim Taskin 

Academic Editor

PLOS ONE